# TimeRewarder: Learning Dense Reward from Passive Videos via Frame-wise Temporal Distance

Yuyang Liu [* 1 2]   Chuan Wen [* 3]   Yihang Hu [1]   Dinesh Jayaraman [4]   Yang Gao [1 2 †]

## Abstract

Designing dense rewards is crucial for reinforcement learning (RL), yet in robotics it often demands extensive manual effort and lacks scalability. One promising solution is to view *task progress* as a dense reward signal, as it quantifies the degree to which actions advance the system toward task completion over time. We present *TimeRewarder*, a simple yet effective reward learning method that derives progress estimation signals from passive videos, including robot demonstrations and human videos, by modeling temporal distances between frame pairs. We then demonstrate how *TimeRewarder* can supply step-wise proxy rewards to guide reinforcement learning. In our comprehensive experiments on ten challenging Meta-World tasks, we show that *TimeRewarder* dramatically improves RL for sparse-reward tasks, achieving nearly perfect success in 9/10 tasks with only 200,000 environment interactions per task. This approach outperforms previous methods and even the manually designed environment dense reward on both the final success rate and sample efficiency. Moreover, we show that *TimeRewarder* can exploit real-world human videos, highlighting its potential as a scalable approach to rich reward signals from diverse video sources. Project page: `timerewarder.github.io`.

## 1. Introduction

Reinforcement learning (RL) has long served as a principal paradigm for robotic skill acquisition (Ibarz et al., 2021;

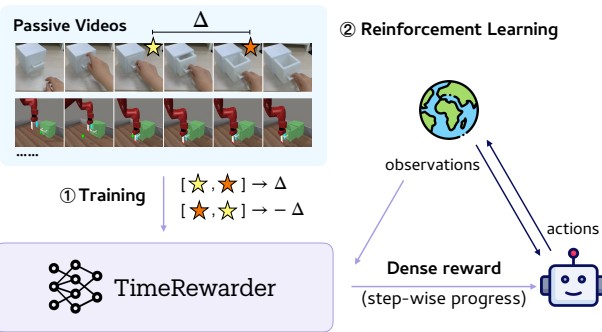

*Figure 1.* Overview of *TimeRewarder*. Mirroring how humans infer task progression by observing others, *TimeRewarder* distills frame-wise temporal distances from expert videos and converts them into dense reward signals, thereby enabling reinforcement learning free of manually engineered rewards or action annotations.

Tang et al., 2025). Yet, many of its most notable successes so far rely highly on carefully designed reward functions that are dense and task-instructive (Cheng et al., 2024; Nai et al., 2025). Designing such high-quality rewards remains labor-intensive, as they often require significant domain expertise, extensive hyperparameter tuning, or privileged access to ground-truth environments, especially for robotic manipulations (Ng et al., 1999a; Levine et al., 2016; Rajeswaran et al., 2017; Roy et al., 2021). These challenges incurred during manual reward design severely constrain the scalability of RL approaches, motivating the development of automated reward learning mechanisms that can alleviate human effort.

Dense reward function design for robotics often exploits explicit prior knowledge of the task's typical progression, which estimates the distance between the current state and task completion, as well as assesses whether the current action contributes to efficient task accomplishment (Todorov, 2004; Levine et al., 2016; Silver et al., 2021). Expert demonstrations provide a natural source of this progression knowledge: the temporal ordering of video frames directly reflects task advancement. Importantly, such signals can be derived even from passive videos, which are easy to obtain and require neither action annotations nor privileged supervision. As a result, automatic reward learning from passive videos can significantly expand the scalability of RL.

Building on this idea, we introduce *TimeRewarder* (Fig-

[1]Institute for Interdisciplinary Information Sciences, Tsinghua University, Beijing, China [2]Shanghai Qi Zhi Institute [3]Shanghai Jiao Tong University [4]University of Pennsylvania. Correspondence to: Yang Gao <gaoyangiiis@mail.tsinghua.edu.cn>, Yuyang Liu <yyliu22@mails.tsinghua.edu.cn>, Chuan Wen <wenchuan@sjtu.edu.cn>.

*Proceedings of the 43rd International Conference on Machine Learning*, Seoul, South Korea. PMLR 306, 2026. Copyright 2026 by the author(s).

ure 1), which comprehends how the task proceeds by learning to predict *temporal distances* between arbitrary frames from action-free expert demonstrations. The temporal distance reflects the *task progress* between two frames: which frame is closer to task completion and by how much. When turning to the RL exploration phase, the predicted progress distances between adjacent frames can naturally serve as dense reward signals. The step-wise reward quantifies exactly how much the agent is advancing or regressing at each moment, guiding the agent toward accomplishing the task by implicitly imitating the expert demonstrations.

We evaluate *TimeRewarder* in the imitation-from-observation setting, where only expert videos are available, without access to expert action labels or dense environment rewards. On 10 Meta-World (Yu et al., 2020) manipulation tasks with 100 demonstrations per task, *TimeRewarder* outperforms all baselines on 9 tasks in terms of both success rate and sample efficiency. These results indicate that the rewards learned by *TimeRewarder* provide informative training signals for RL: they capture fine-grained task progress, distinguish unproductive or suboptimal behaviors, and generalize to agent-generated trajectories beyond the expert data. As a result, *TimeRewarder* enables effective policy learning from action-free videos, reducing reliance on carefully engineered reward functions. More broadly, this work offers a simple and scalable instantiation of how temporal structure in passive videos can be leveraged for reward learning in watch-to-act settings.

## 2. Related Works

Previous work has explored methods of learning from observation-only demonstrations, providing agents with task-relevant supervision when environmental rewards are sparse or inaccessible.

**Action Recovery.** Model-based approaches (Nair et al., 2017; Torabi et al., 2018a; Pathak et al., 2018; Edwards et al., 2019; Radosavovic et al., 2021; Fan et al., 2022; Baker et al., 2022; Liu et al., 2022; Ramos et al., 2023) aim to recover missing actions in expert demonstrations by learning inverse dynamics models from online interaction data, followed by behavioral cloning on the recovered action labels. In practice, training reliable inverse dynamics models requires large amounts of transition data and typically involves iterative online data collection to sufficiently cover the state distribution of expert demonstrations. These requirements make the overall pipeline data-intensive and sensitive to exploration quality, which can limit its applicability in real-world robotic settings.

**Inverse RL.** Instead of explicitly recovering actions for behavior cloning, Inverse RL aims to build reward functions from expert demonstrations (and online interactions if needed) to guide policy updates within a standard RL paradigm. Trajectory-matching methods (Dadashi et al., 2020; Yang et al., 2019; Jaegle et al., 2021; Chen et al., 2021; Haldar et al., 2023; Liu et al., 2024) measure rollout–expert similarity as a reward signal, while adversarial imitation learning (Ho & Ermon, 2016; Torabi et al., 2018b) trains a discriminator to distinguish agent from expert transitions. With the advance of generative models, some recent works (Escontrela et al., 2023; Huang et al., 2024) train video generation models and take the likelihood of rollout frames produced by this model as the reward. Despite the progress of these methods, they face challenges such as high online computational cost (Haldar et al., 2023; Escontrela et al., 2023), training instability (Ho & Ermon, 2016), or reward hacking (Escontrela et al., 2023).

**Progress-based Reward Learning.** Within inverse RL, some methods define proxy rewards by exploiting the temporal structure of demonstrations, where the ordering of frames along a trajectory provides an implicit measure of *task progress*. TCC (Dwibedi et al., 2019) enforces cycle-consistency in time for correspondence, while Arrow of Time (Wei et al., 2018) exploits temporal irreversibility for representation learning. TCN (Sermanet et al., 2018) pulls temporally adjacent frames together in the latent visual representation space while pushing distant ones apart, though it enforces only coarse temporal consistency and produces non-locally smooth representations (Ma et al., 2022). Building on this, VIP (Ma et al., 2022) estimates frame–goal distances using implicit time-contrastive learning. However, we found this objective unbounded and difficult to optimize reliably (Detailed proof in Appendix. A.3). GVL (Ma et al., 2024) uses vision-language models to infer temporal orders from shuffled frames, yet we observed that its reliance on inconsistent VLM outputs limits its effectiveness in building reward functions. Rank2Reward (Yang et al., 2024) learns to predict the temporal order of adjacent frame pairs, providing lightweight local rewards; PROGRESSOR (Ayalew et al., 2024) considers triples of frames to estimate the relative position of an intermediate frame between start and goal states. However, Rank2Reward predicts only the relative ordering of frame pairs without modeling explicit temporal distance, whereas PROGRESSOR focuses solely on forward progression and relies on a more complex objective.

In contrast, *TimeRewarder* directly estimates *frame-wise temporal distances* without goal conditioning. This self-consistent objective fully exploits temporal structure, leading to stable optimization and robust performance.

## 3. Preliminaries

### 3.1. Learning from Action-free Demonstrations

We study the problem of learning policies from action-free expert demonstrations. Specifically, the agent has access to a dataset of expert RGB videos besides an environment

to interact with. We resolve the problem from the RL perspective, by deriving a proxy reward from the action-free demonstrations, which is used to guide downstream policy optimization.

Formally, we consider an agent interacting with a finite-horizon Markov Decision Process $(\mathcal{S}, \mathcal{A}, \mathcal{P}, \mathcal{R}, \gamma, T)$, where $\mathcal{S}$ is the state space, $\mathcal{A}$ the action space, $\mathcal{P}$ the transition dynamics, $\mathcal{R}$ the reward function, $\gamma$ the discount factor, and $T$ the horizon. We assume the agent can not access states $s_t \in \mathcal{S}$ directly, but only high-dimensional visual observations $o_t \in \mathcal{O}$ in the form of RGB images. Moreover, the environmental reward function $\mathcal{R}$ provides only sparse binary success signals indicating whether the task is completed or not, which is easily obtainable via human annotation or vision-language model API.

Such sparse signals are far from enough for guiding efficient policy optimization. To overcome this, we derive a proxy reward from the expert data, hoping that the agent can receive instructive learning signals even when the environmental reward remains zero during exploration. We denote the expert dataset as $D^e = \{\tau_i^e\}$, where $\tau^e = (o_1^e, o_2^e, \ldots, o_T^e)$ represents observation trajectories. The goal is to recover a proxy reward function $\hat{\mathcal{R}}$ from $D^e$, such that a policy $\pi^{\hat{\mathcal{R}}}$ trained on this reward:

$$\pi^{\hat{\mathcal{R}}} = \arg\max_\pi \mathbb{E}\left[\sum_{t=1}^T \gamma^{t-1}\hat{\mathcal{R}}(o_t, o_{t+1})\right] \qquad (1)$$

can successfully accomplish the task.

### 3.2. Progress-based Reward Design

Since the agent's ultimate objective is to reach a goal state, the distance to task completion can be interpreted as a measure of task progress, which can inform reward design. This idea is closely related to potential-based reward shaping (Ng et al., 1999b), where the reward at each transition is defined as the change in a potential function $V(o)$ that measures the progress-to-go from $o$ toward the goal:

$$r_t = \hat{\mathcal{R}}(o_t, o_{t+1}) = V(o_t) - \gamma V(o_{t+1}). \qquad (2)$$

Such progress-based proxy rewards offer two primary benefits: (1) *Generality*: Task progress is a high-level signal implicitly encoded in expert demonstrations, avoiding the need for hand-crafted reward design. (2) *Action-free learning*: Progress can be inferred directly from passive video data, without requiring access to action labels. These properties yield dense and temporally consistent feedback, enabling policy learning from action-free video demonstrations.

## 4. Method

We introduce *TimeRewarder*, a framework that derives dense proxy rewards for downstream RL by estimating task progress from action-free expert videos. The central idea is to model progress as a *temporal distance prediction problem*: learning to estimate the temporal distance between two observations in a trajectory. In this section, we (1) formalize the construction and training of *TimeRewarder*, (2) present its application in deriving reward functions for RL, and (3) provide a theoretical justification demonstrating that temporal distance aligns naturally with task progress.

### 4.1. Training with Frame-wise Temporal Distance

We train *TimeRewarder*, a progress model $F_\theta : \mathcal{O} \times \mathcal{O} \to \mathbb{R}$, on expert demonstrations $D^e$. The model learns to predict the normalized temporal distance between two ordered frames $(o_u^e, o_v^e)$, providing a dense signal of task progress. As shown in Figure 2 (a), given two frames $(o_u^e, o_v^e)$ from an expert trajectory, their normalized temporal distance is computed as:

$$d_{uv} = \frac{v - u}{T - 1} \in [-1, 1], 1 \le u, v \le T, \qquad (3)$$

so that $F_\theta$ is trained in a self-supervised manner, taking two ordered frames and predicting the relative temporal distance between them.

To be effective as a reward signal, $F_\theta$ must satisfy two key principles: (1) **Suboptimality Awareness** — generalize beyond expert data and assign lower scores to suboptimal behaviors which are unseen in $D^e$; (2) **Fine-grained Temporal Resolution** — capture fine-grained progress, particularly between adjacent steps.

For *suboptimal awareness*, *TimeRewarder* naturally realizes **Implicit Negative Sampling**: the frame indices $u$ and $v$ in (3) can appear in either forward or backward order, so the normalized temporal distance $d_{uv}$ ranges from $-1$ to $1$. A positive value indicates forward progression toward the goal, while a negative value indicates backward progression, naturally corresponding to movement away from task completion, simulating suboptimal or incorrect behaviors. This formulation imposes an antisymmetric structure on the learning objective, thereby discouraging trivial memorization shortcuts (Ma et al., 2024).

As for *fine-grained temporal resolution*, we aim to enhance the model's ability to recognize progress at the step level, i.e., between adjacent frames, so that the learned metric can provide informative step-wise rewards. To this end, we introduce **Weighted Pair Sampling**: sample frame pairs with a bias shorter intervals while still ensuring coverage of longer horizons. Concretely, for a frame pair $(o_u^e, o_v^e)$ with temporal interval $\Delta = |v - u|$, we sample $\Delta$ with probability

$$P(\Delta) \propto \frac{1}{\Delta}, \quad \Delta \in \{1, \ldots, T-1\}. \qquad (4)$$

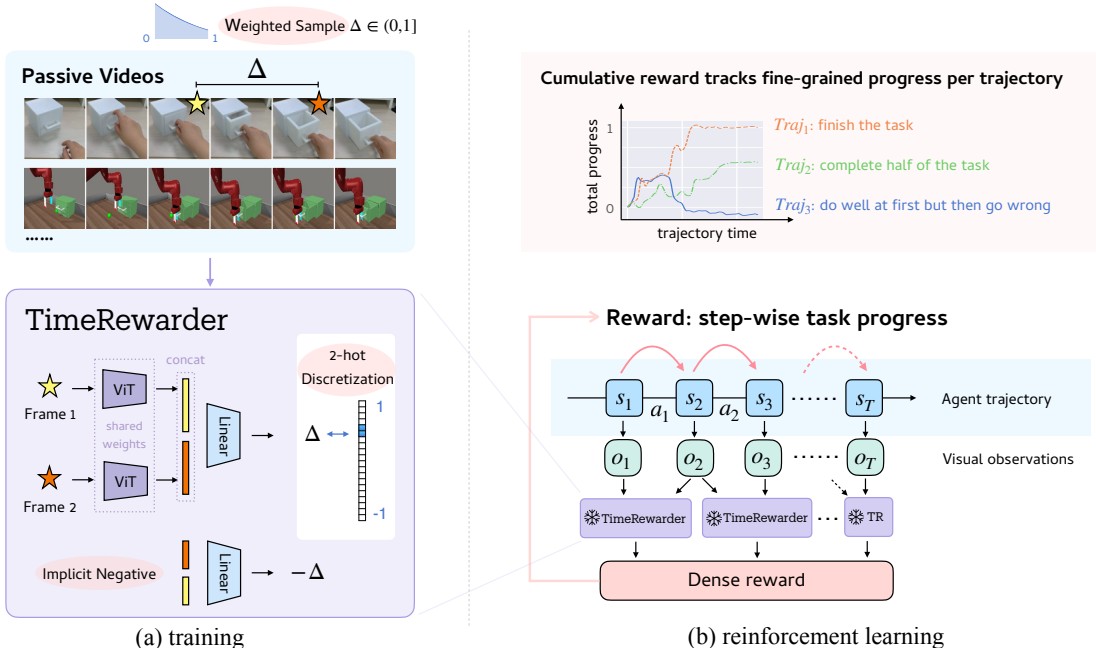

(a) training           (b) reinforcement learning

*Figure 2. **TimeRewarder** framework. **TimeRewarder** learns step-wise dense rewards from passive videos by modeling intrinsic temporal distances, enabling robust progress scoring that assigns high values to states reflecting task advancement, while penalizing suboptimal actions lacking meaningful contribution to task progression, thereby facilitating effective policy learning.*

This simple sampling scheme emphasizes fine-grained local differences while retaining the ability to capture broader temporal dependencies.

Additionally, to ensure numerical stability and maintain accuracy during the optimization process, we employ **Two-hot Discretization** (Wang et al., 2024) to discretize the scalar temporal distance $d_{uv} \in [-1, 1]$. Specifically, the target range $[-1, 1]$ is uniformly partitioned into $K$ bins (we set $K = 20$ by default). For a given $d_{uv}$, we compute a soft two-hot distribution $\mathbf{y}_{uv} = \Phi(d_{uv}) \in \mathbb{R}^K$ that assigns non-zero mass only to the two nearest bins. The progress model $F_\theta$ outputs a logit vector $\hat{\mathbf{y}}_{uv} = F_\theta(o_u^e, o_v^e) \in \mathbb{R}^K$, and the training objective is the cross-entropy loss:

$$\min_\theta \; \mathbb{E}\big[ -\mathbf{y}_{uv}^\top \log \mathrm{softmax}(\hat{\mathbf{y}}_{uv})\big]. \quad (5)$$

Through this training, $F_\theta$ learns a robust notion of temporal progress inside any ordered frame pairs from purely observational passive video data.

### 4.2. Policy Learning with Temporal Distance Reward

Then, we utilize $F_\theta$ to provide dense proxy rewards for RL. As illustrated in Fig. 2 (b), for each policy rollout, **TimeRewarder** computes adjacent frame distances as step-wise rewards ($\Phi^{-1}$ is the inverse transform of $\Phi$, mapping a two-hot vector back to the scaler):

$$r_{\mathrm{TR}}(o_t, o_{t+1}) = \hat{d}_{t,t+1} = \Phi^{-1}\big[F_\theta(o_t, o_{t+1})\big] \in [-1, 1], \quad (6)$$

where the output logits of $F_\theta$ have been converted back to a scalar value.

During policy optimization, we combine this progress-based dense reward with a sparse success signal:

$$r_t = r_{\mathrm{TR}}(o_t, o_{t+1}) + \alpha \cdot r_{\mathrm{success}}(o_t), \quad (7)$$

where $r_{\mathrm{success}} : \mathcal{O} \to \{0, 1\}$ is a binary success indicator (1 if successful, 0 otherwise), and $\alpha \geq 0$ is a weighting factor used to align the scales of the dense and sparse reward components, preventing either term from dominating due to differences in magnitude. While the method itself is compatible with a fixed $\alpha$, in practice, we choose $\alpha$ adaptively in our experiments to account for scale differences across reward formulations used by different methods. Appendix A.4.1 provides implementation details and shows that performance is not sensitive to the specific choice of $\alpha$.

Although $F_\theta$ is trained solely on expert trajectories, its design ensures natural generalization to diverse behaviors. Suboptimal behaviors—such as stalls, loops, or regressions—receive lower or even negative rewards, while meaningful partial progress is still recognized and positively rewarded. This graded, step-wise feedback provides informative signals for exploration, guiding the agent to recover from failures and make constructive progress toward task completion. Together with the sparse success signal, this mechanism allows **TimeRewarder** to produce dense and informative rewards throughout training, which underlies its empirical effectiveness demonstrated in Section 5.

## 4.3. Theoretical Justification

We provide a theoretical justification for our motivation that the task progress in expert videos can be formalized in terms of temporal distance. Consider a fully observable Markov Decision Process (MDP), where the true state $s \in \mathcal{S}$ captures all task-relevant information (e.g., object poses, velocities, and gripper status). Given a goal state $s_g$, we define a sparse reward with a per-step penalty:

$$r(s) = \begin{cases} -1, & s \neq s_g \\ 0, & s = s_g \end{cases} \tag{8}$$

and assume deterministic transitions $s' = f(s, a)$.

Under this setting, the optimal value function satisfies the Bellman optimality equation:

$$\mathcal{V}_\gamma^*(s) = r(s) + \gamma \max_a \mathcal{V}_\gamma^*(f(s, a)). \tag{9}$$

For any expert trajectory $\tau^e = (s_1^e, \ldots, s_T^e)$ generated by the optimal policy, the value of each visited state admits a closed-form expression:

$$\mathcal{V}_\gamma^*(s_t^e) = -\sum_{k=t}^{T-1} \gamma^{k-t}, \quad \mathcal{V}^*(s_g) = 0. \tag{10}$$

Therefore, $\mathcal{V}^*(s)$ is a monotonic transformation of the remaining time-to-go $T - t$. This naturally suggests defining a potential function:

$$V(s_t^e) = -\mathcal{V}_{\gamma=1}^*(s_t^e) = T - t, \tag{11}$$

which maps each state to its temporal distance to the goal.

**Connection to visual RL.** To bridge the above formulation with visual observations, we assume the underlying state can be approximately reconstructed from observations, i.e., $s \approx \phi(o)$. This requires that each observation uniquely identifies the task phase, thereby ruling out cyclic trajectories that revisit visually indistinguishable states. In practice, however, visual RL operates in a Partially Observable MDP (POMDP), where this assumption can be violated due to *visual aliasing*, particularly in back-and-forth motions. In such cases, disambiguating different visits would require history-dependent information (e.g., a loop counter or velocity), which is not available from single-frame observations.

For example, consider a trajectory $(o_0, o_1, o_2, o_3, o_1, o_g)$, where the first occurrence of $o_1$ corresponds to the hand reaching to open a drawer, and the second corresponds to retracting after placing an object. In the underlying state space, these correspond to distinct states $s_1$ and $s_4$, with true temporal distances of 4 and 1 steps to the goal, respectively, thus preserving monotonic progress. However, when restricted to single-frame observations, both states collapse to the same $o_1$, leading a learned temporal distance model

$F_\theta$ to produce an averaged estimate (e.g., 2.5), which may locally violate monotonicity.

To resolve this in practical deployments where back-and-forth motions are frequent, *TimeRewarder* can be extended by replacing single-frame inputs with observation histories (e.g., concatenating a time window of frames $(o_{t-k}, \ldots, o_t)$). By incorporating history, the input once again approximates the true Markovian state vector, effectively disambiguating the temporally distinct visits to $o_1$ and recovering the accurate progress distances of 4 and 1.

## 5. Experiments

In this section, we assess the performance of *TimeRewarder*. We present the experiment setup, evaluate *TimeRewarder* against baselines, and do ablations of its key components.

### 5.1. Experiment Setup

**Evaluation Benchmark.** We evaluate *TimeRewarder* and other methods on ten challenging Meta-World (Yu et al., 2020) manipulation tasks (see Appendix A.1 for details). For each task, we provide 100 action-free expert videos generated by Meta-World's scripted policies. These videos serve as the training data for reward learning methods; for ILfO methods, rewards are computed with respect to the most closely aligned demonstration. For three tasks, we further consider a cross-domain setting where only one in-domain expert video is provided per task, supplemented with 20 real-world human demonstration videos.

**Implementation Details.** We use a CLIP-pretrained ViT-B (Radford et al., 2021; Dosovitskiy et al., 2021) as the visual backbone of *TimeRewarder*. During training, frame pairs are independently encoded, concatenated, and passed through a linear layer to predict discretized temporal distances. Both the ViT-B encoder and linear layer are trainable. For RL, *TimeRewarder* is integrated with DrQ-v2 (Yarats et al., 2021), and the whole network is frozen, providing dense step-wise rewards from adjacent observation frames. See Appendix A.5.3 for hyperparameters.

**Baselines.** We compare **T**imeRewarder against eight baselines, grouped into three categories:

*Progress-Based Reward Learning:* **PROGRESSOR** (Ayalew et al., 2024) fits a Gaussian model to estimate relative frame positions between initial and goal as rewards; **Rank2Reward** (Yang et al., 2024) estimates temporal rank between frames as rewards; and **VIP** (Ma et al., 2022) trains an implicit value model to estimate task progress of each frame given a goal image. Goal frames sampled from expert videos are provided to PROGRESSOR and VIP, following their original settings. For a fair comparison, *TimeRewarder* and all three baselines adopt CLIP-pretrained ViT-B as

the same vision backbone. For VIP, we additionally report results with its default ResNet-34 backbone in Appendix A.4.2. Results with a from-scratch ViT-B backbone, ablating visual pretraining, are reported in Appendix A.4.3.

*Imitation Learning from Observations:* **GAIfO** (Torabi et al., 2018b), **OT** (Papagiannis & Li, 2022), and **ADS** (Liu et al., 2024) compute rewards online by comparing rollouts to expert videos. GAIfO uses a discriminator, OT applies Wasserstein distance via Optimal Transport (Villani et al., 2009), and ADS extends OT with curriculum scheduling on the discount factor to better handle progress-dependent tasks. Comparison against other OT-based methods, including TemporalOT (Fu et al., 2024) and ORCA (Huey et al., 2025), is shown in Appendix A.4.4.

*Privileged Methods:* For reference, we also report results of policies with access to privileged information: **BC** (Bain & Sammut, 1995) trains a behavior cloning policy with expert actions, and **Environment reward** uses Meta-World's ground-truth dense reward.

For the seven baselines involving reinforcement learning (except **BC**), we uniformly adopt DrQ-v2 (Yarats et al., 2021) as the underlying RL algorithm for fair comparison.

### 5.2. Performance of TimeRewarder

We address the following five questions to structure our experimental results and analysis, to demonstrate the performance achieved by *TimeRewarder* against the baselines.

**Question 1.** *Does TimeRewarder provide correct task progress for unseen success trajectories rather than relying on memorization?*

A well-shaped reward should encourage successful rollouts with monotonic progress, even when trajectories differ from training demonstrations in object positions or motion paths. We test *TimeRewarder* and progress-based reward baselines under the Value-Order Correlation (VOC) metric (Ma et al., 2024), which evaluates the alignment between predicted values and temporal order (+1 for perfect monotonicity increasing, 0 for no correlation, −1 for inverse). Specifically, we train *TimeRewarder* and VIP on 100 expert demonstrations and test them on 100 held-out expert videos. To further strengthen the empirical comparison, we introduce GVL (Ma et al., 2024) implemented

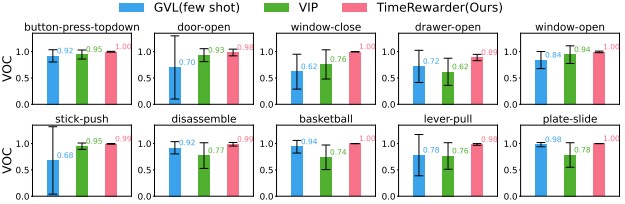

*Figure 3.* Value–Order Correlation (VOC) on held-out expert videos. Higher is better.

with `Gemini-1.5-Pro` (Team et al., 2024) as an additional baseline, where we follow its few-shot setting by giving 5 expert videos as context and another 5 for testing, where 32 frames are uniformly sampled from each video. Rank2Reward and PROGRESSOR are excluded because they encode progress as dense rewards rather than potential-based value functions. As shown in Figure 3, *TimeRewarder* consistently achieves the highest VOC scores, confirming its strong temporal coherence and generalization to unseen trajectories.

**Question 2.** *Can TimeRewarder identify suboptimal behavior in rollout trajectories?*

Reward models trained only on successful demonstrations inevitably face out-of-distribution transitions during RL exploration, where they may misinterpret them by either over-estimating failures or undervaluing successes. We select one representative successful (`traj1`) and one failed (`traj2`) trajectory from two tasks, and visualize the progress estimates of *TimeRewarder* against baselines in Figure 4.

In the *basketball* task, where `traj2` grasps but never lifts the ball, VIP ignores partial progress and PROGRESSOR saturates after grasping, while Rank2Reward and *TimeRewarder* cleanly capture half-success and then separates completion from failure. In the *window-open* task, where `traj2` mimics opening motions midair without contacting the handle, VIP and Rank2Reward are misled by visual similarity, PROGRESSOR gives spurious early spikes which can mislead exploration, whereas *TimeRewarder* increases values only upon meaningful interaction. Rank2Reward, limited to pairwise orderings, fails to produce consistent distinctions. These comparative results demonstrate *TimeRewarder*'s unique capacity for temporally coherent and causally grounded feedback under distribution shift—significantly outperforming previous methods in distinguishing productive from unproductive behaviors.

**Question 3.** *Does TimeRewarder learn structured representations that generalize across demonstrations and rollouts?*

To better understand *TimeRewarder*'s robustness, we examine the structure of the feature space induced by the learned progress model $F_\theta$. Concretely, $F_\theta$ encodes each observation using a shared encoder and predicts temporal distance based on their feature difference (not the concatenated feature as in our default setting). We visualize these per-observation features along trajectories in Figure 5.

Across both training and held-out expert demonstrations, *TimeRewarder* learns smooth and well-structured representations that evolve consistently with task progress, indicating strong temporal coherence and generalization. Importantly, this structure extends to RL rollouts: successful trajectories follow a coherent progression in representation space toward the goal, whereas failed trajectories deviate from this structure, reflecting their lack of meaningful progress.

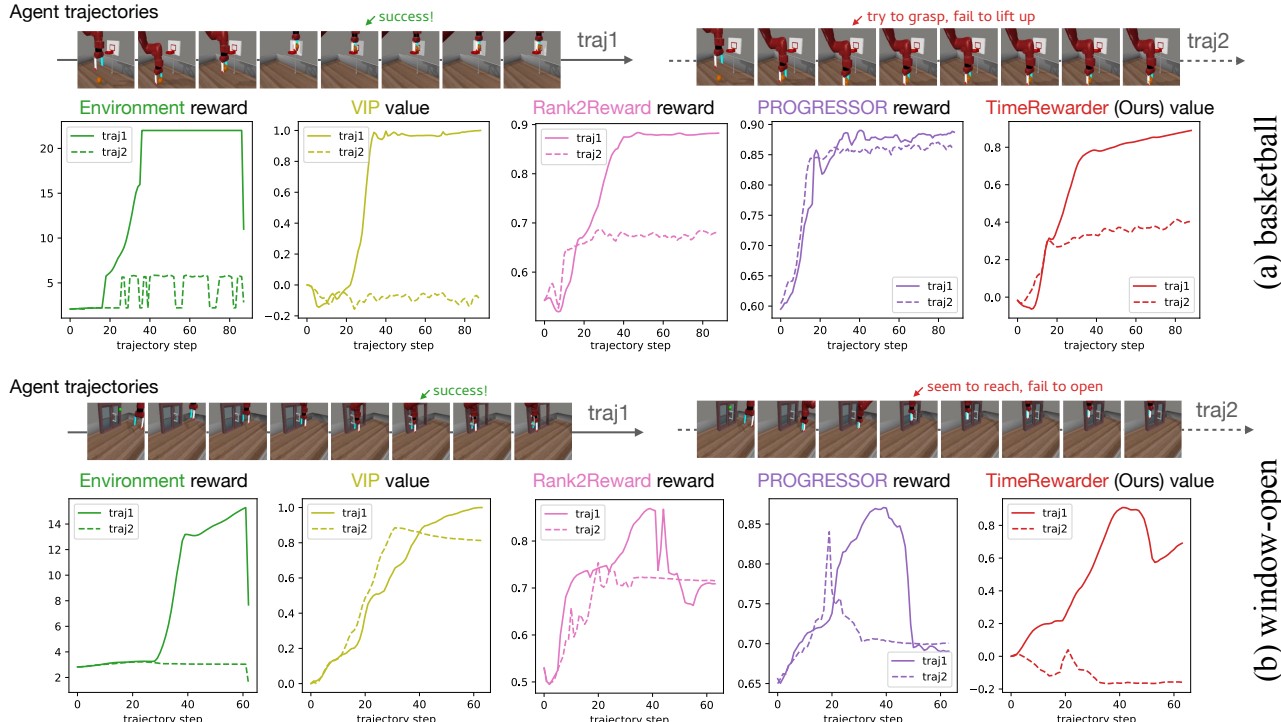

*Figure 4.* Reward/value curves on successful (`traj1`) vs. failed (`traj2`) rollouts for two tasks. ***TimeRewarder*** and VIP output *values* (cumulative progress), Rank2Reward and PROGRESSOR output stepwise *rewards*, all curves reflect *progress estimation*. ***TimeRewarder*** provides the most informative and temporally coherent feedback.

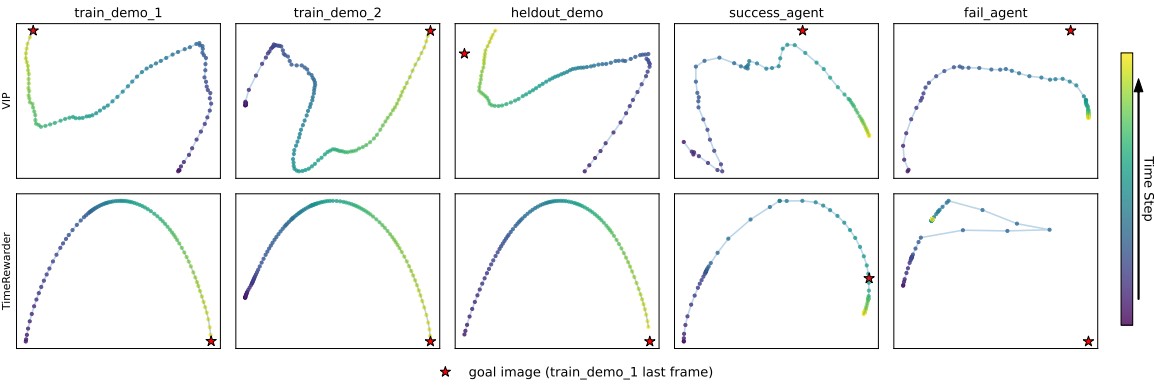

*Figure 5.* PCA visualization of VIP and TimeRewarder representations on trajectories of the `window-open` task. `train_demo_1` and `train_demo_2` are expert demonstrations from the training set; `heldout_demo` is a different expert demonstration used to evaluate generalization. `success_agent` and `fail_agent` are trajectories from RL rollouts same as `traj1` and `traj2` in Figure 4(b).

In contrast, VIP representations exhibit weaker structure. While they capture coarse trends on training demonstrations, they are less smooth and degrade on held-out trajectories, indicating limited generalization. Moreover, VIP fails to clearly separate successful and failed rollouts, suggesting a tendency toward representation collapse under distribution shift. We attribute this limitation in part to its reliance on a predefined goal image: when observations lie far from the specified goal, the learned representation becomes less informative. This issue similarly affects other goal-conditioned approaches such as PROGRESSOR.

**Question 4.** *Can TimeRewarder improve reinforcement learning performance?*

We present the downstream RL performance of ***TimeRewarder*** against baselines in Figure 6. Specifically, we implement DrQ-v2 with rewards summed up from the proxy rewards produced by these methods and the environmental binary success signals, similar to Eq.(7). We see that ***TimeRewarder*** attains the highest final success rate and the greatest sample efficiency on 9 of 10 tasks. Remarkably, ***TimeRewarder*** also outperforms policies trained with dense **Environment reward** on 9 tasks, which

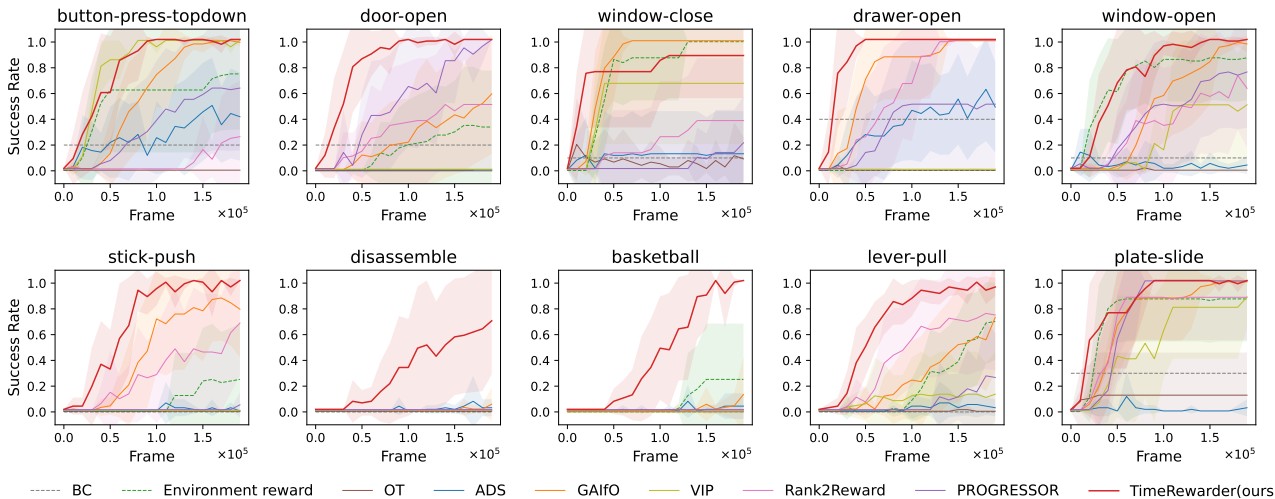

Figure 6. Performance of reinforcement learning with sparse environment success signals and dense proxy rewards from each method. Curves show mean ± s.d. over eight seeds. Dashed lines indicate reference settings of behavior cloning (BC) and environment dense reward supervision.

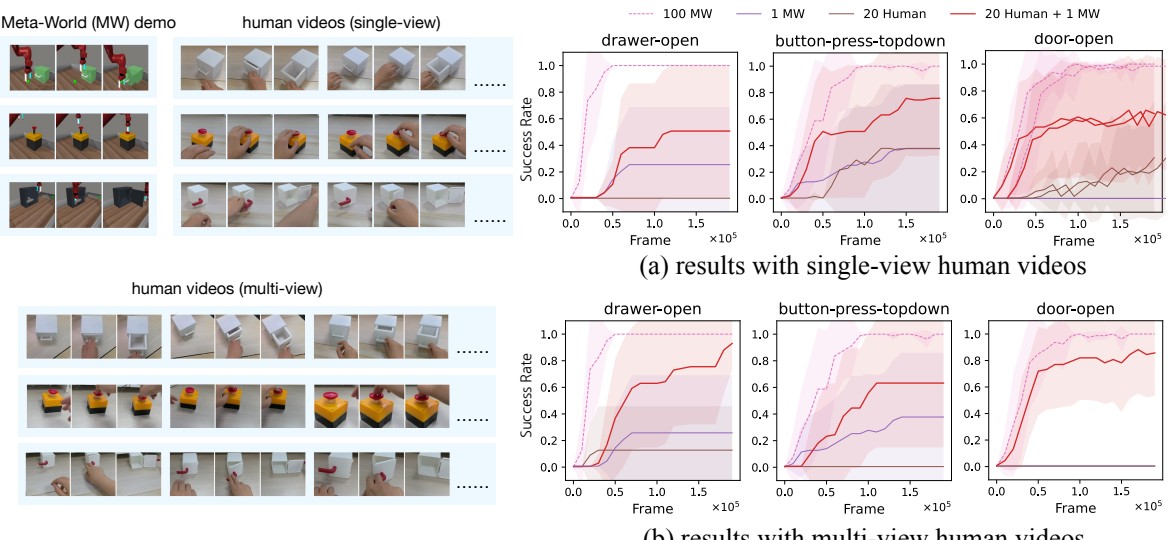

Figure 7. Cross-domain reward learning. *TimeRewarder* improves performance by leveraging 20 unlabeled human videos alongside only 1 in-domain Meta-World demonstration per task, demonstrating its ability to utilize cross-domain visual data. Curves show mean ± s.d. over eight seeds.

is commonly treated as an upper bound. We attribute this to the fact that hand-crafted dense rewards in Meta-World often provide limited feedback on pre-contact progress that is critical for effective control, whereas *TimeRewarder* captures fine-grained incremental progress from demonstration videos. This yields a smoother and more informative learning signal without requiring task-specific design. Additional experiments without environment success signals are provided in Appendix A.4.5.

**Question 5.** *Can TimeRewarder generalize across different domains and even embodiments?*

To test cross-domain generalization, we choose 3 tasks and build their corresponding copies in real-world settings. For

these 3 tasks we collect 20 human demonstrations individually under each of the following 2 camera settings: fixed viewpoint or varying viewpoints. Such cross-domain videos, together with a single in-domain expert video from the original Meta-World environment, are then provided to *TimeRewarder* for reward learning and downstream RL. As shown in Figure 7, training on either human-only (brown) or Meta-World-only (purple) data yields low success rates, but combining them (red) substantially improves performance. These results highlight the ability of *TimeRewarder* to leverage cross-domain, unlabeled video data for reward learning, even when in-domain supervision is scarce. The full set of human videos is shown in Appendix A.2.

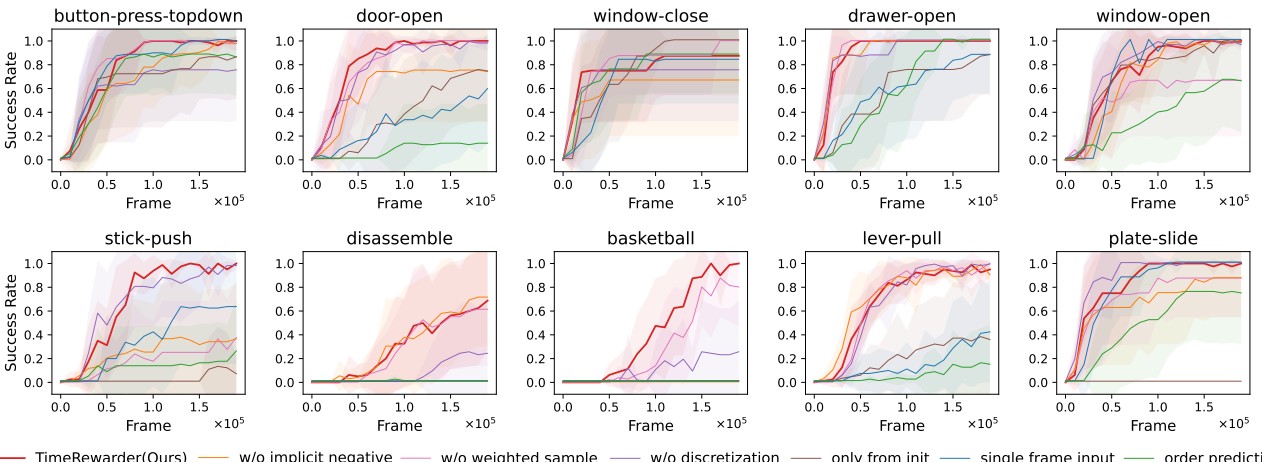

*Figure 8.* Ablation study results. Curves show mean ± s.d. over eight seeds.

## 5.3. Ablation Studies

In Figure 8, we evaluate the contribution of each component in Section 4.1 through controlled removals:

**Effect of Implicit Negative Sampling.** Implicit negative sampling enforces *suboptimal awareness* by treating reverse-ordered frame pairs as implicit negatives, simulating failures during training. Removing it and predicting only forward progress $\in [0, 1]$ causes sharp drops in *stick-push* and *basketball* (orange line), where failed grasps are common and should not be interpreted as partial progress. Without negatives, the model overestimates such failures as partial success. PROGRESSOR exhibits a similar trend (Figure 6), suggesting that explicitly accounting for failure-like transitions can be beneficial for these types of sparse-reward manipulation tasks. Additional ablation on adding negative sampling to PROGRESSOR is shown in Appendix A.4.8.

**Effect of Weighted Sampling.** Weighted sampling biases training toward shorter temporal intervals while still retaining coverage over longer horizons, allowing the model to capture subtle, temporally localized state changes. Replacing it with uniform sampling leads to reduced performance on *stick-push* and *window-open* (pink line), tasks that require precise, temporally localized interactions. Without increased emphasis on adjacent frames, subtle but task-relevant state changes become harder to resolve, resulting in less informative progress estimates that provide weaker learning signals for precise control.

**Effect of Discretization.** Two-hot discretization ensures *numerical stability* and sharp progress boundaries by binning temporal distances. Direct regression causes large drops in *basketball* and *disassemble* (purple line), where long setup phases are followed by brief decisive actions (e.g., lifting the ball or ring). Direct regression smooths over these moments, failing to distinguish success from near-success, while discretization preserves sharp transitions and provides

stronger completion incentives. We further analyze the discretization bin number $K$ in Appendix A.4.6, showing that performance is largely insensitive to $K$.

We additionally evaluate three alternative design choices (details in Appendix A.5.1): (1) **only from init** measures progress solely relative to the initial frame; (2) **single-frame input** predicts progress for individual frames rather than relative progress between frame pairs; and (3) **order prediction**, inspired by GVL (Ma et al., 2024), reconstructs sequences from shuffled frames. All three variants underperform *TimeRewarder*. The first two exhibit limited temporal expressiveness, while the third introduces additional complexity without yielding consistent performance gains.

## 6. Conclusion

We present *TimeRewarder*, a simple yet effective method that produces dense instructive rewards by learning to predict temporal distances from action-free expert videos. This approach captures fine-grained task progress, naturally accounts for suboptimal behaviors, and provides informative step-wise feedback for RL. Experiments on diverse robotic manipulation tasks demonstrate that *TimeRewarder* not only outperforms prior reward learning methods but also surpasses environment-supplied dense rewards, in terms of both success rate and sample efficiency. Additionally, *TimeRewarder* demonstrates successful cross-domain learning ability by leveraging real-world human videos to improve policy learning, when in-domain data is limited.

In summary, *TimeRewarder* provides a promising direction for reducing reliance on manual reward engineering. Although current limitations emerge on tasks with frequent back-and-forth motions, we expect them to be addressed by future hierarchical or memory-augmented progress models, so that scalable "watch-to-act" skill acquisition from in-the-wild video becomes truly attainable.

## Impact Statement

This work advances reward learning by offering a practical and scalable way to exploit temporal progress signals from action-free videos, addressing a long-standing challenge in effectively modeling and utilizing such information. Rather than relying on hand-crafted reward functions, *TimeRewarder* learns dense rewards directly from demonstrations, reducing manual engineering while achieving stronger empirical performance than manually designed environment rewards. Even with limited in-domain data, our results show that heterogeneous video sources, including real-world human demonstrations, can meaningfully benefit policy learning, highlighting the robustness and generality of the approach. Overall, *TimeRewarder* represents a step toward more accessible and scalable robot skill acquisition from video. While current limitations arise in tasks with frequent back-and-forth motions, we expect future hierarchical or memory-augmented progress models to address these challenges, further enabling scalable watch-to-act learning from in-the-wild videos.

This paper presents work whose goal is to advance the field of Machine Learning. There are many potential societal consequences of our work, none of which we feel must be specifically highlighted here.

## Acknowledgment

This research was conducted with the support of the Shanghai Qi Zhi Institute and the Tsinghua University Dushi Program. Funding and support for this work were also provided by the Tsinghua University - Keystone Electrical (Zhejiang) Co., Ltd Joint Research Center for Embodied Multimodal Artificial Intelligence (JCEMAI). Additionally, we would like to extend our thanks to the Xiongan AI Institute.

We thank Jiacheng You (IIIS, Tsinghua University) for helpful discussions. We also sincerely thank the anonymous reviewers for their thoughtful comments and constructive feedback, which helped improve this paper.

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

# A. Appendix

## A.1. Tasks for Evaluation

In this paper, we experiment with the following 10 tasks from the Meta-World suite (Yu et al., 2020):

1. **Button press topdown:** to press a button from the top.

2. **Door open:** to open a cabinet door with a handle.

3. **Window close:** to close a sliding window with a handle.

4. **Drawer open:** to open a cabinet drawer with a handle.

5. **Window open:** to open a sliding window with a handle.

6. **Stick push:** to pick up a stick and push a kettle with the stick.

7. **Disassemble:** to pick and remove a nut from a peg.

8. **Basketball:** to pick up a basketball and dump it into a basket.

9. **Lever pull:** to pull a lever up 90 degrees.

10. **Plate slide:** to push a plate into the goal area.

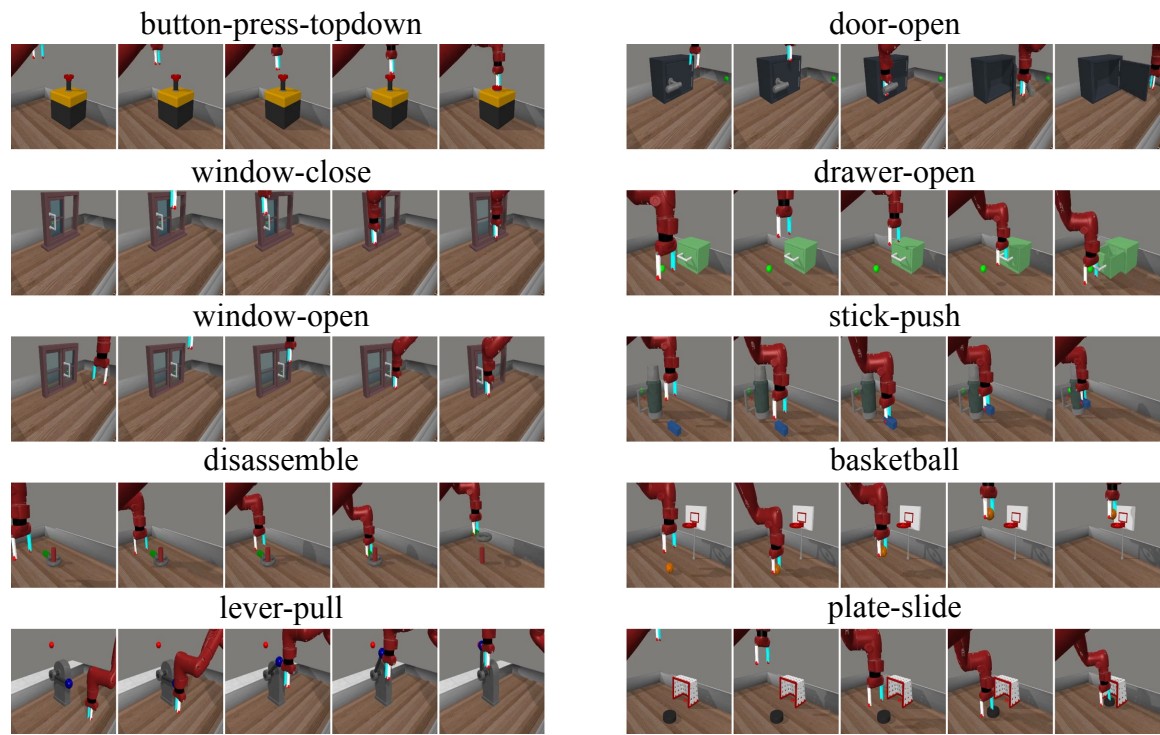

*Figure 9.* Meta-World tasks used in our paper.

## A.2. Human Video Datasets for Cross-Domain Experiments

This section presents the complete set of human videos used in the cross-domain experiments across three tasks. Each task includes 20 videos recorded in *single-view* (fixed viewpoint) and 20 videos recorded in *multi-view* (varying viewpoints) conditions. These videos differ from the robot setting in embodiment and background, and contain no action or state annotations.

The full set of videos for each task in both conditions is shown in Figure 10 and Figure 11.

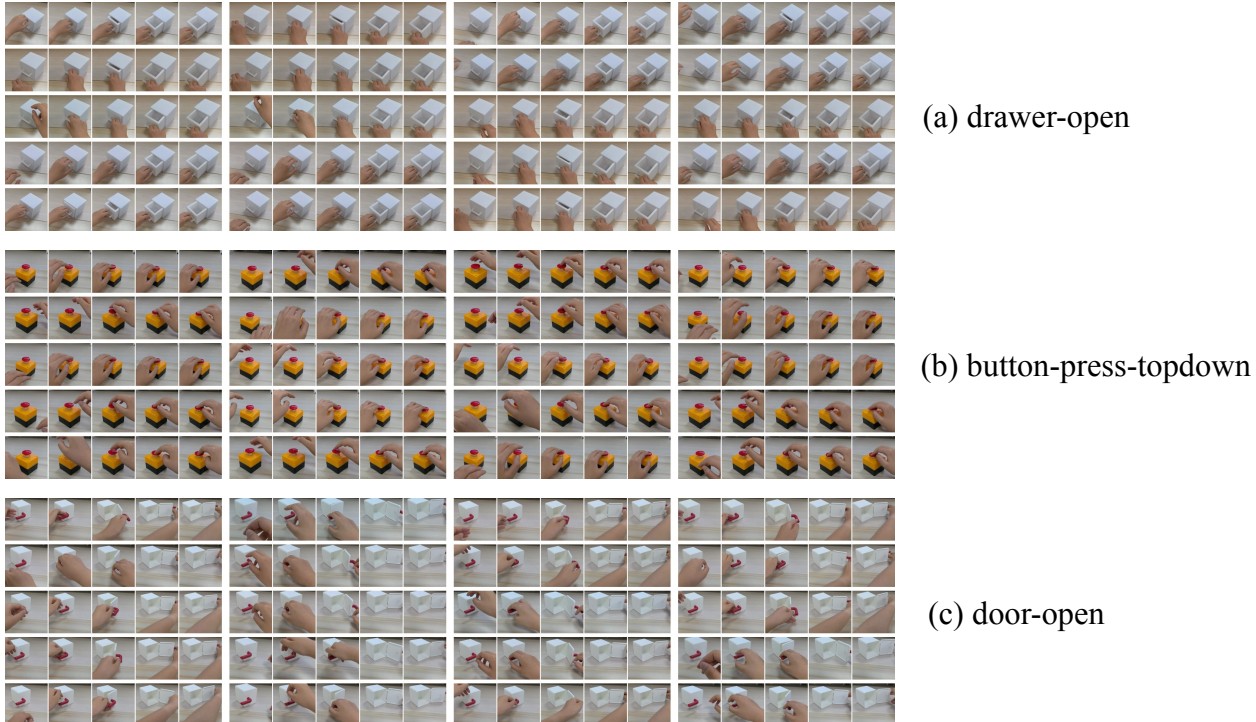

*Figure 10.* Complete set of human videos recorded in the *single-view* condition for each of the three tasks. Each task includes 20 videos captured from a fixed viewpoint.

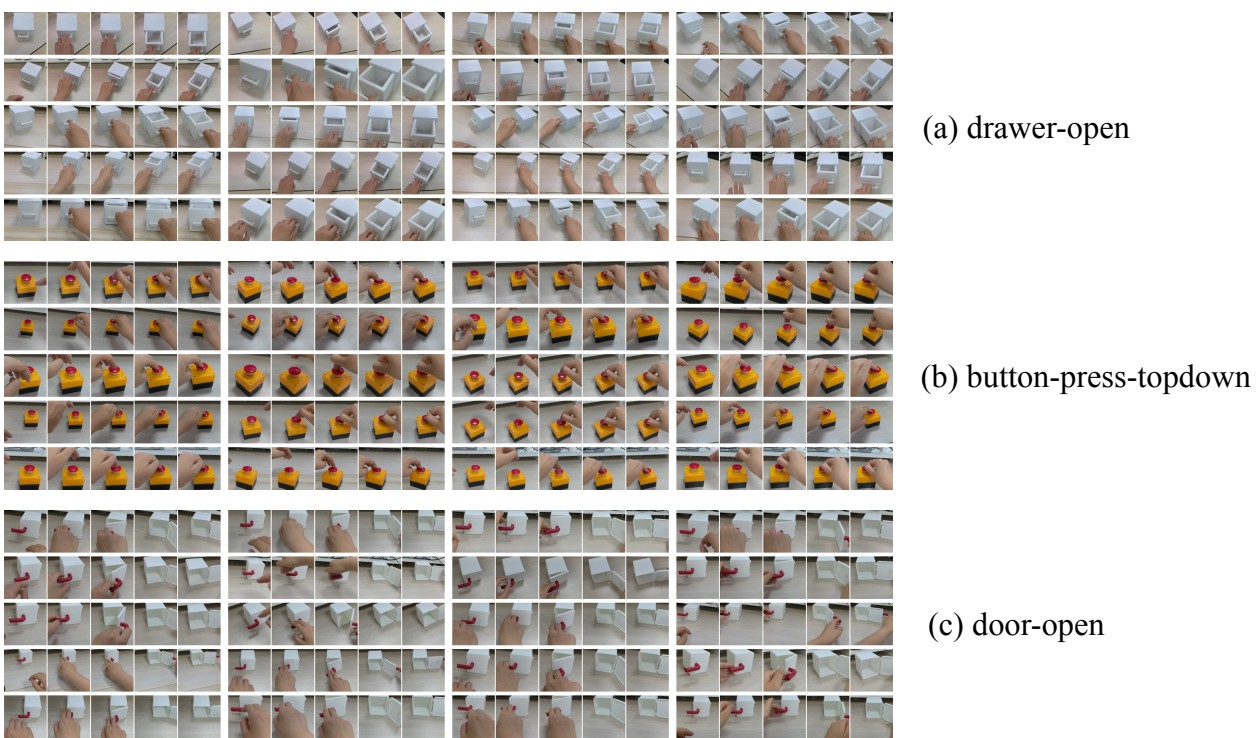

*Figure 11.* Complete set of human videos recorded in the *multi-view* condition for each of the three tasks. Each task includes 20 videos captured from varying viewpoints.

## A.3. Additional Proof: Boundlessness of VIP Objective

We note that the objective equation in the VIP paper (Ma et al., 2022), Eq.(6), is inconsistent with its pseudocode in Appendix D.3. Following the pseudocode (consistent with the official codebase), the sign before $\gamma$ is reversed. Under this formulation, we show that the VIP loss has no lower bound and admits degenerate solutions with unbounded representations:

In VIP Section 4.1, $\tilde{\delta}_g(o) = \mathbb{I}(o == g) - 1$, so $\tilde{\delta}_g(o) = 0$ if and only if $o = g$ instead of $o_{next} = g$. (VIP codebase: "this is always -1" in `data_loaders.py`.)

Consider an $N+1$-state chain-like MDP with initial state $o_0$ and goal $g = o_N$. For a full-trajectory batch $\{o_0, o_1, \ldots, o_{N-1}\}$, all samples share the same $o_0$. Note that $V_{o_t} = -\|\phi(o_t) - \phi(g)\|$, and we have

$$\mathcal{L} = (1-\gamma)(-V_{o_0}) + \log\left(\frac{1}{N}\sum_{t=0}^{N-1}\exp\left(-(\tilde{\delta}_g(o) + \gamma V_{o_{t+1}} - V_{o_t})\right)\right)$$

$$= (1-\gamma)\|\phi(o_0) - \phi(g)\| + \log\left(\frac{1}{N}\sum_{t=0}^{N-1}\exp\left(\gamma\|\phi(o_{t+1}) - \phi(g)\| - \|\phi(o_t) - \phi(g)\|\right)\right) + 1.$$

Let

$$\|\phi(o_t) - \phi(g)\| = c\frac{1-\gamma^{N-t}}{1-\gamma}$$

$(c > 0)$, we have

$$\gamma\|\phi(o_{t+1}) - \phi(g)\| - \|\phi(o_t) - \phi(g)\| = c\frac{\gamma(1-\gamma^{N-(t+1)}) - (1-\gamma^{N-1})}{1-\gamma} = c\frac{\gamma - 1}{1-\gamma} = -c,$$

Thus

$$\mathcal{L} = (1-\gamma)c\frac{1-\gamma^N}{1-\gamma} + \log\left(\frac{1}{N}\sum_{t=0}^{N-1}\exp(-c)\right) + 1 = c(1-\gamma^N) - c + 1 = -\gamma^N c + 1.$$

Obviously $\lim_{c\to+\infty}\mathcal{L} = -\infty$.

In practice, we indeed observed that when effective representations encoding progress are learned, the solution approximately resembles the Bellman solution scaled by a constant, appearing to "converge." Yet this outcome is not guaranteed, which is why we noted in the paper that this objective is "difficult to optimize reliably."

## A.4. Additional Experiment Results

### A.4.1. CHOICE OF REWARD COMBINATION FACTOR

In (7) we use a weight constant $\alpha$ to align the scales of the dense and sparse components so that neither term dominates purely due to magnitude differences, since different methods produce reward functions on different scales. In practice, we use an adaptive way to choose $\alpha$, which is to calculate the maximum value of the dense reward for the first few (100) trajectories, and then set $\alpha$ to ten times this maximum value.

Figure 12 further shows that the performance of *TimeRewarder* is not sensitive to the value of $\alpha$.

### A.4.2. VIP BACKBONES

In the main experiments (Figure 6), all baselines, including Rank2Reward, PROGRESSOR, and VIP, use the same CLIP-pretrained ViT-B backbone as *TimeRewarder* to ensure a fair comparison. Since VIP originally adopts a ResNet-34 backbone in its official implementation, we additionally report VIP results with its default ResNet-34 backbone in this appendix.

As shown in Figure 13, the relative performance of VIP with ResNet-34 and ViT-B varies across tasks: ResNet-34 performs better on some tasks, while ViT-B achieves higher performance on others. Nevertheless, regardless of the backbone choice,

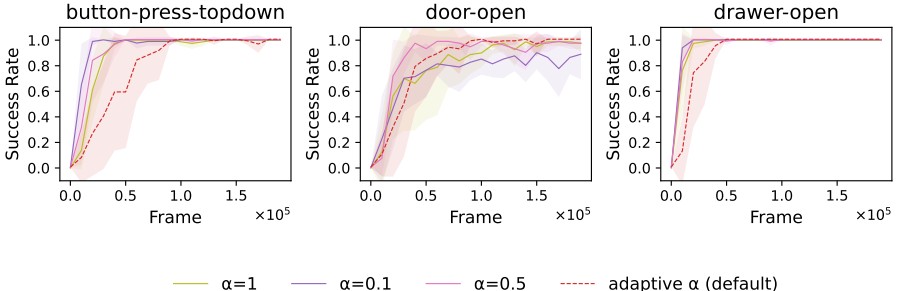

Figure 12. *TimeRewarder*'s performance with different $\alpha$. Curves show mean $\pm$ s.d. over eight seeds.

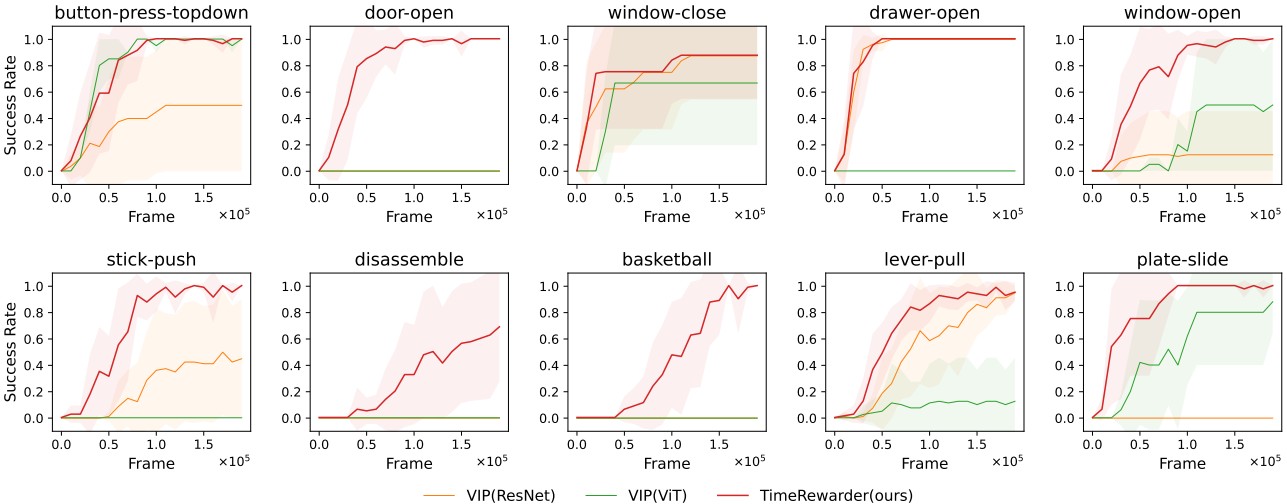

Figure 13. VIP performance with ResNet34 vs. ViT backbones across tasks. The results show that *TimeRewarder* outperforms VIP regardless of the backbone used. All methods are evaluated with reinforcement learning using sparse environment success signals and dense proxy rewards. Curves show the mean $\pm$ s.d. over eight seeds.

*TimeRewarder* consistently outperforms VIP across all tasks. This indicates that although the backbone architecture can affect VIP's performance on individual tasks, the performance gap between VIP and *TimeRewarder* cannot be attributed to backbone selection.

All experiments follow the same protocol described in Section 5.1, where methods are evaluated on the same 10 Meta-World manipulation tasks with sparse binary success rewards.

### A.4.3. TRAINING FROM SCRATCH

Figure 14 presents an ablation study in which a ViT-B model trained from scratch is used as the visual backbone, replacing the CLIP-pretrained ViT-B employed in the main experiments. We compare TimeRewarder against the baseline methods Rank2Reward and PROGRESSOR. As expected, the absolute performance decreases due to the weaker visual representations. Nevertheless, the overall trends persist, and TimeRewarder continues to outperform the baseline methods.

### A.4.4. MORE OT-BASED BASELINES

Figure 15 compares *TimeRewarder* with two more optimal transport(OT)-based imitation learning from observations methods, including TemporalOT (Fu et al., 2024) and ORCA (Huey et al., 2025).

TemporalOT does not outperform OT or ADS, consistent with our observation that OT-based methods rely on near-identical initial states; the masking mechanism in TemporalOT further exacerbates this, leading to near-zero success when trajectories differ. This matches observations in ORCA.

ORCA can outperform ADS, but its max-based alignment allows frame skipping without penalty, and its multiplicative

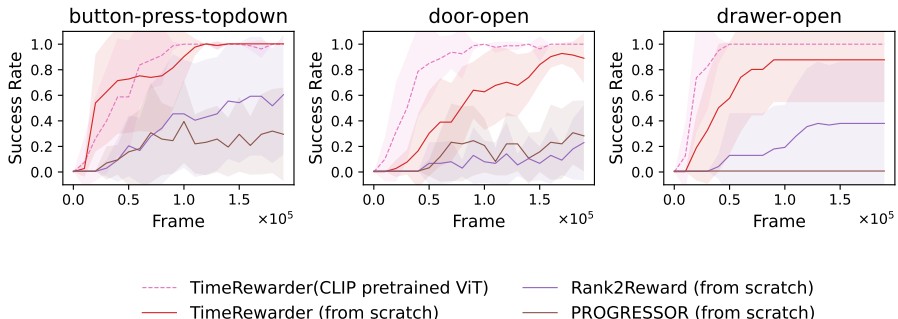

*Figure 14.* Ablation study with from-the-scratch ViT-B as the vision backbone. Curves show mean $\pm$ s.d. over eight seeds.

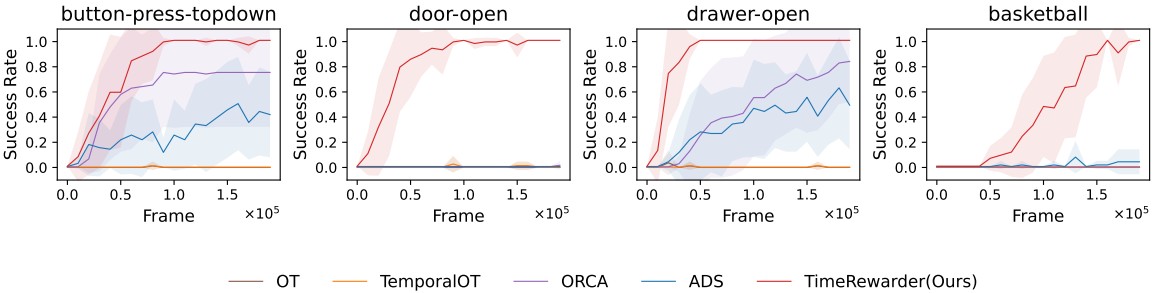

*Figure 15.* Comparison with TemporalOT and ORCA. Curves show mean $\pm$ s.d. over eight seeds.

structure penalizes long-horizon matches, resulting in strong task-dependent variance, as reflected in the table.

In contrast, TimeRewarder learns signed temporal distances over all frame pairs, avoiding both compounding errors and frame-skipping issues, and achieves more stable performance across tasks.

### A.4.5. RL WITH ONLY PROXY REWARD

Compared to Figure 6, Figure 16 presents the results when the environment's sparse reward is entirely removed, relying solely on the learned proxy reward. Additionally, we include results for the ILfO baseline **BCO** (Torabi et al., 2018a; Baker et al., 2022). Under the constraint of extremely short training (only 200,000 frames), no successes are achieved. However, by the end, the agent has started making progress and completing part of the task, though not the full goal.

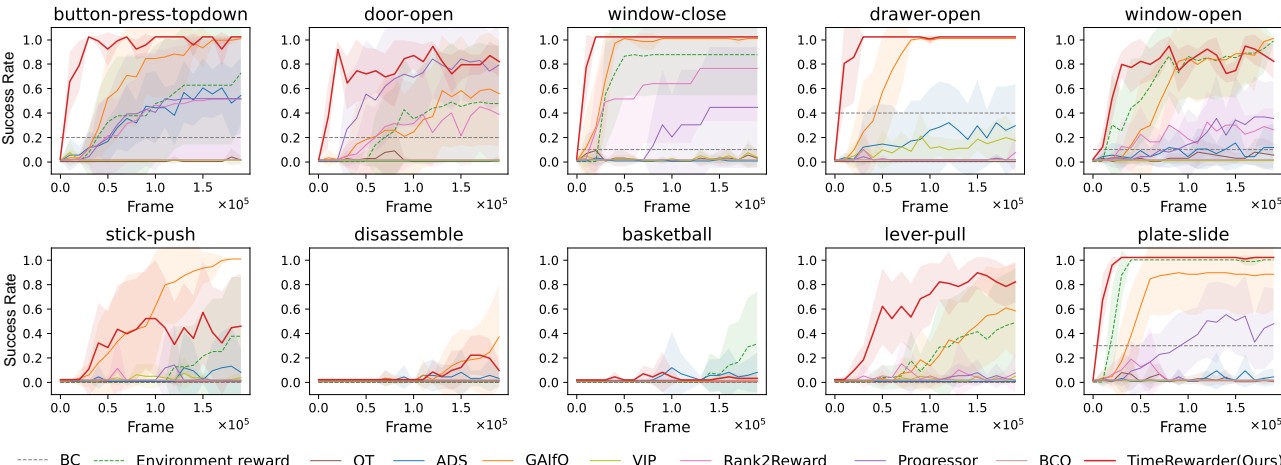

*Figure 16.* Reinforcement learning without sparse reward. Curves show mean $\pm$ s.d. over eight seeds. Dashed lines indicate behavior cloning (BC) and environment dense reward supervision.

### A.4.6. CHOICE OF DISCRETIZATION BINS

Figure 17 shows the performance of *TimeRewarder* with different bin number $K$ in 2-hot discretization. While the default $K = 20$ represents a practical tradeoff between reward precision and optimization stability, the performance is not very sensitive to the value of $K$.

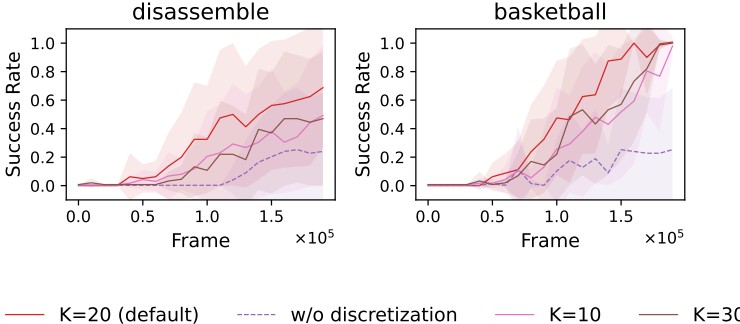

*Figure 17. TimeRewarder*'s performance with different bin number $K$ in 2-hot discretization. Curves show mean $\pm$ s.d. over eight seeds.

### A.4.7. ADDITIONAL CROSS-DOMAIN EXPERIMENTS

In the main experiment results, with 100 Meta-World demonstrations, performance is already strong. Adding 20 human videos yields similar performance but slightly improves sample efficiency by further broadening the coverage. The results are shown in Figure 18 .

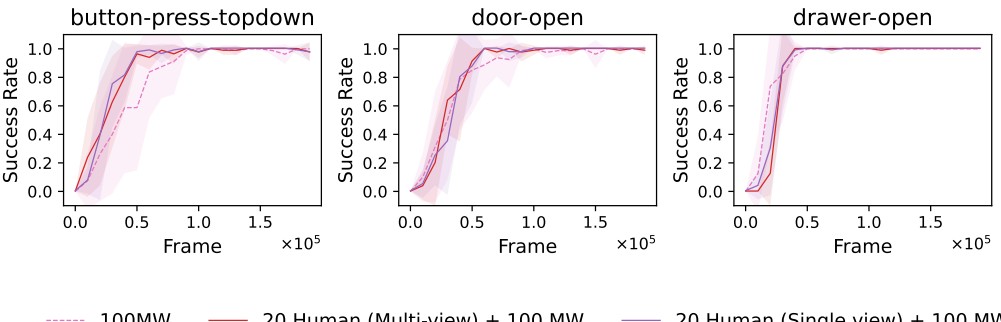

*Figure 18.* Cross-domain experiment results of adding 20 human videos to 100 Meta-World demonstrations. Curves show mean $\pm$ s.d. over eight seeds.

### A.4.8. ABLATION STUDY OF PROGRESSOR

To further understand the role of structural design choices, we conduct an ablation by introducing negative sampling into PROGRESSOR's original training objective. The goal is to examine whether enriching the supervision signal can close the performance gap to *TimeRewarder*.

PROGRESSOR originally selects three consecutive frames $(o_1, o_2, o_3)$ and predicts the relative position of the middle frame $o_2$ within a local temporal window$(o_1, o_3)$. This formulation only models forward progress and does not naturally support antisymmetric temporal reasoning. To test whether this limitation can be mitigated, we augment the training with negative sampling (predict $o_1$ relative to $(o_2, O_3)$), enabling the model to also compare a given frame against non-adjacent or temporally reversed counterparts.

As shown in Figure 15 (right panel), this modification leads to noticeable improvements on tasks such as `window-close` and `drawer-open`, where local temporal structure is relatively consistent and easier to exploit. However, the improvement is not uniform across tasks. In more challenging scenarios such as `basketball` and `stick-push`, the ablated PROGRESSOR still significantly underperforms compared to *TimeRewarder*.

We attribute this gap to a structural limitation: even with negative sampling, PROGRESSOR does not explicitly learn an

antisymmetric representation over frame pairs. The new formulation shifts the prediction range from $[0, 1]$ to $[-T, 1]$, which does not enforce a consistent notion of bidirectional temporal distance, making it difficult to represent both progression and regression in a unified embedding space. In contrast, *TimeRewarder* directly models frame-wise signed temporal distances, which naturally encode such antisymmetry.

These results suggest that while data augmentation strategies such as negative sampling can partially improve PROGRESSOR, the core architectural constraint remains the main bottleneck. The persistent gap across several tasks indicates that performance differences are primarily driven by representation structure rather than optimization details alone.

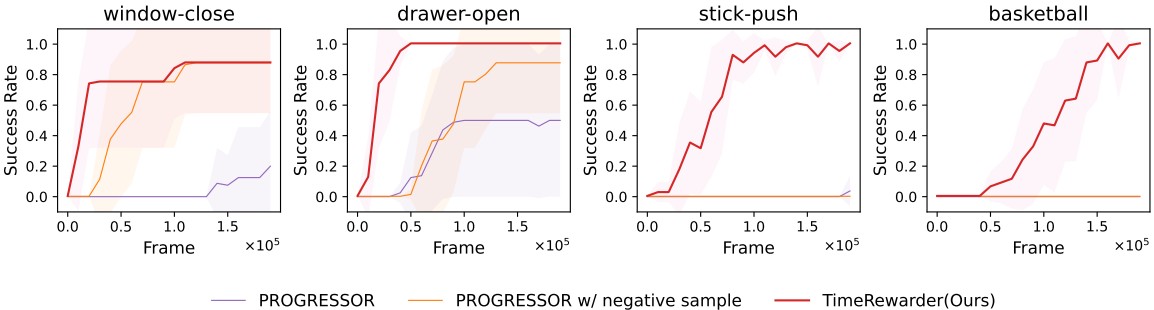

*Figure 19.* Ablation study of PROGRESSOR. Curves show mean $\pm$ s.d. over eight seeds. While adding negative sampling significantly improves performance on `window-close` and `drawer-open`, tasks like `basketball` and `stick-push` still underperform compared to TimeRewarder, since PROGRESSOR's structure does not naturally support antisymmetric (inverse) representation modeling.

### A.4.9. ADDITIONAL SIMULATOR

We have completed the training and evaluation for the `PushCube-v1` task in the ManiSkill3 (Tao et al., 2025) suite. We maintained a strictly identical setup to our Meta-World experiments: we used the first 100 trajectories from the official ManiSkill demonstrations for all the reward models (including TimeRewarder, VIP, Progressor, and Rank2Reward) and ILFO (OT, GAIfO) methods, and subsequently trained the downstream RL policies based purely on these rewards. *TimeRewarder*'s strong performance on ManiSkill provides further evidence that this signal generalizes across different simulator dynamics and visual rendering.

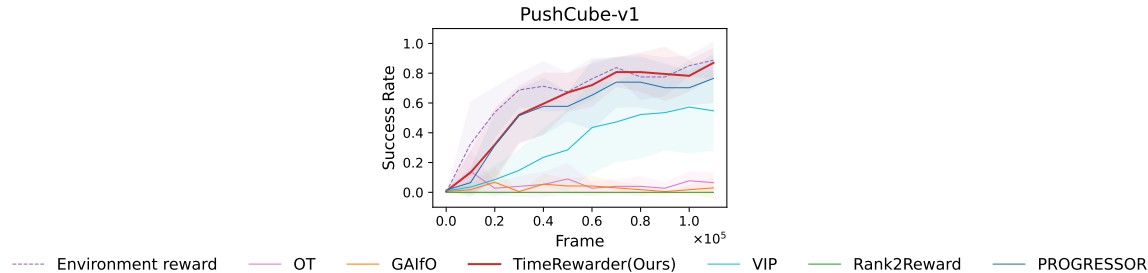

*Figure 20.* RL performance on `PushCube-v1` task in ManiSkill3 suite. Curves show mean $\pm$ s.d. over eight seeds.

### A.5. Implementation details

#### A.5.1. ALTERNATIVE TEMPORAL MODELING APPROACHES

In our main method, *TimeRewarder* does temporal modeling through predicting the relative progress between two frames in a video. We also examined other three temporal modeling approaches as following.

**(1) only from init.** Considering the distribution shift, predicting the progress from each frame in an agent's rollout trajectory to a goal image derived from another expert trajectory may not be suitable. In addition to the goal frame, a natural choice is to use the initial frame as an anchor, which captures the positions of objects in the environment. In this context, when sampling frame pairs from expert trajectories, instead of randomly selecting any two frames, we fix the first frame as the

initial frame. We then predict the progress within the range of [0, 1], while adhering to the three methodological components in *TimeRewarder*.

**(2) single frame input.** The simplest method to capture temporal information in a video is to directly predict the normalized temporal position (ranging from [0,1]) of each individual frame. In contrast to *TimeRewarder*, we use only one frame as input for our reward model instead of two. We uniformly sample the frame and apply the discretization technique.

**(3) order prediction.** Our order prediction setting is inspired by the setup of GVL (Ma et al., 2024). During training, we uniformly sample $n = 32$ frames from each expert video and apply a random permutation. The model is trained to recover the original ordering using a cross-entropy loss over permutation positions. At test time, we input an agent trajectory and predict a score for each frame reflecting its position in the estimated order. The model architecture mirrors that of *TimeRewarder*, but replaces the temporal regression head with a frame-wise classifier for permutation indices. Specifically, the predicted scalar values are normalized between $[-1, 1]$.

**Reward computation:** For all three methods mentioned above, the prediction of the reward model reflects the progress of an agent's trajectory at each time step. These scores are then utilized as potentials in a potential-based reward formulation. Consequently, the reward for each step is defined as the forward difference between successive predicted values.

### A.5.2. DEMONSTRATION COLLECTION FOR META-WORLD

To better approximate in-the-wild video data, we collected Meta-World demonstrations under a deliberately diverse initialization protocol. Rather than using the default narrow initialization range, where both agents and experts begin from nearly identical configurations, we expanded the initial state space to cover a broad variety of robot and object positions. This leads to demonstrations with much greater appearance diversity and prevents agent trajectories from being trivially aligned to demonstrations at the pixel level.

This choice also explains the results in Figure 6, where occupation-matching methods such as Optimal Transport (OT) and its extension ADS perform poorly. With the default narrow initialization, agents and experts share similar starting conditions, allowing OT and ADS to exploit appearance-level shortcuts when aligning trajectories. Once the initialization range is broadened, these shortcuts disappear, and the assumptions underpinning OT and ADS no longer hold, leading to degraded performance.

Crucially, this setting more faithfully reflects real-world conditions, where demonstrations and agent experiences seldom begin from the same initial states. It therefore underscores the importance of methods like *TimeRewarder* that extract robust progress signals rather than depending on superficial appearance matching.

### A.5.3. HYPERPARAMETERS

For reward learning, we use a ViT-B/16 backbone. Frame features are extracted, concatenated into a 1024-dimensional vector, and projected through a linear layer into 20 discretized bins. Training data is augmented to 10,000 pairs per epoch. The hyperparameters are summarized in Table 1.

*Table 1.* Reward model hyperparameters.

| Config | Value |
|---|---|
| Backbone | ViT-B/16 |
| Feature dimension | $1024(512 \times 2)$ |
| Output bins $K$ | 20 (two-hot discretization) |
| Training pairs per epoch | 10,000 |
| Epochs | 100 |
| Warm-up epochs | 5 |
| Batch size | 16 |
| Accumulation steps | 1 |
| Optimizer | Adam |
| Learning rate | $2 \times 10^{-5}$ |

We equip all the methods with the same underlying RL algorithm, DrQ-v2 (Yarats et al., 2021). The hyperparameters are

listed in Table 2.

*Table 2.* RL hyperparameters.

| Config | Value |
| --- | --- |
| Replay buffer capacity | 150000 |
| $n$-step returns | 3 |
| Mini-batch size | 512 |
| Discount | 0.99 |
| Optimizer | Adam |
| Learning rate | $10^{-4}$ |
| Critic Q-function soft-update rate $\tau$ | 0.005 |
| Hidden dimension | 1024 |
| Exploration noise | $\mathcal{N}(0, 0.4)$ |
| Policy noise | $\text{clip}(\mathcal{N}(0, 0.1), -0.3, 0.3)$ |
| Delayed policy update | 1 |

