# OpenReview forum: "TimeRewarder: Learning Dense Reward from Passive Videos via Frame-wise Temporal Distance"
_ICML.cc/2026/Conference — ICML 2026 spotlight_

### Official Review · Reviewer_KVj4 · 2026-02-18

**Soundness:** 2
**Presentation:** 3
**Significance:** 3
**Originality:** 3
**Overall Recommendation:** 5
**Confidence:** 4

**Summary:**

The paper proposes a method to extract dense rewards from videos, termed TimeRewarder. The idea is that task progress can be expressed by modeling a distance metric between frames, where the frames are typically obtained from expert action-free videos. To that end, a “progress model” is trained to predict a scalar value representing the temporal distance between frames, where the value is the [-1, 1] normalized index-distance between the frames (> 0 indicates forward progress and backwards otherwise).
The method is demonstrated on the sparse-reward-based MetaWorld benchmark, demonstrating favorable performance over numerous baselines.

**Compliance With Llm Reviewing Policy:**

Affirmed.

**Final Justification:**

The authors have addressed all my concerns and made a significant effort during the rebuttal period to strengthen the paper, which I truly appreciate. I raised my score and recommend accepting this paper.

**Key Questions For Authors:**

* My main key question is related to the weakness I raised regarding “evidence”. A main challenge of working with videos is the discrepancy between the frame-rate-per-second (FPS) of videos and the control frequency of the robot (whether real-world or simulated). High-frequency control typically leads to small differences between frames and larger otherwise. How does your method address that? I believe that some of the results based on human videos are affected by that. It is not entirely clear if and how we can apply this method with general-purpose video pre-training and specific robotic benchmarks.

* I do not understand how $\Delta$ is sampled (Eq .4) and would like a clarification for that.

* Do the authors plan to provide open-source code/checkpoints?

* Overall, I like the idea of this work and I appreciate the extensive analysis; however, it is hard for me to get convinced that it works beyond MetaWorld, especially in more challenging benchmarks that include multi-object manipulation.

**Limitations:**

Limitations are not discussed beyond the one sentence in the conclusion section. Given that the only benchmark is MetaWorld, and all the insights are tailored for this environment, I believe a broader discussion of limitations can help assess the method’s utilization for other benchmarks/tasks.

**Strengths And Weaknesses:**

**Strengths**:
* Paper is well-motivated, the sparse-reward and reward engineering are challenging issues, and generating dense rewards to guide visual-based RL methods is justified.
* The reward model shows some potential of utilizing human videos instead of just in-domain data.
* Extensive analysis of the proposed reward for different cases with various related baselines.
* Presentation: aside for some minor comments (see below), the paper is clear and easy-to-follow.
* Significance: while the method is only demonstrated on one simulated benchmark, it does preset progress in self-supervised reward design for RL.
* Extensive appendix.

**Weaknesses**:
* Evidence: the main weakness in my opinion is that the method is only demonstrated on one simulated benchmark, and it is hard to assess whether there is enough evidence for the claims in the paper. Mainly, different environment/task may include behaviors that do not align with the assumptions in this work like optimality (e.g., depending on the quality of expert data, what you define as “forward progress” may not necessarily be apparent in the data, like jittery gripper for example).

* Limitations: very light discussion of limitations in the conclusion section. Given the extensive analysis of the results, I would expect a more elaborate discussion of limitations (beyond one sentence).

**Minor**:

* The term “passive videos” is extensively used throughout the manuscript, but its meaning is never defined. What do the authors mean by “passive”? Is it just unlabeled action-free videos?

* Lines 124-125: “which is easily obtainable via human annotation…” - I would tone down the claims regarding the obtainability of sparse rewards from human annotations or VLMs, as the former requires human effort while the latter can not always be reliable. I understand the need to justify that task success is something that does not require a lot of burden, but it is not always so simple, especially from visual inputs.

* Eq. 6 - if $\Phi^{-1}$ refers to the conversion of logits to scalars, it should be explicitly mentioned (“where $\Phi^{-1}$ is the…”). Otherwise, please explain what it is.

---

> ### Author Rebuttal · Authors · 2026-03-28
>
> # Weakness1 & Question1: Challenges about control frequency and suboptimal experts
> We thank the reviewer for raising these important points. We agree that evaluating only in simulation is a limitation, and we will clarify this in the revision. More importantly, we emphasize that our goal is to study how temporal signals from videos can be leveraged in a general and scalable way, rather than being tied to a specific benchmark. In fact, our experiments span multiple Meta-World tasks with diverse dynamics, where prior methods exhibit different failure modes, while our method remains consistently effective.
>
> Regarding the mismatch between video FPS and control frequency, this does not pose a fundamental issue in our formulation. Temporal distances are normalized, and during training we sample arbitrary pairs of frames from the same trajectory, rather than relying only on adjacent frames. This exposes the model to a wide range of temporal gaps, so the learned progress estimation is not tied to a specific frame rate. Moreover, in downstream RL, rewards are evaluated within the agent’s own rollouts, without requiring alignment to the original expert videos. Therefore, even if the control frequency differs from the training data, the method only needs to maintain consistent progress estimation across the agent’s own rollouts, rather than matching a fixed external reference. In particular, rewards are directly comparable between different rollouts of the same agent, without requiring online alignment to expert videos (as in Rank2Reward or PROGRESSOR). This makes the approach robust to FPS differences between training videos and deployment.
>
> We also agree that expert demonstrations may be noisy or suboptimal (e.g., jittery motions). Compared to methods such as Rank2Reward, which rely on pairwise ordering, our approach leverages all frame pairs within trajectories, providing richer supervision and reducing sensitivity to such local noise in practice.
>
> We will clarify these points and better discuss real-world applicability and limitations in the revision.
>
> # Weakness2 & Question4: Limitations
> We thank the reviewer for this valuable feedback. We agree that the discussion of limitations in the current version is brief, and we will expand it in the revision.
>
> In particular, our method assumes roughly monotonic progress and does not explicitly model long-horizon dependencies or multi-object interactions. More complex settings, such as multi-object manipulation or tasks with frequent reversals, may violate these assumptions and are left as future work, potentially requiring hierarchical or memory-augmented mechanisms.
> At the same time, even in these more complex tasks, segments with approximately monotonic progress remain, and the core idea of TimeRewarder, learning dense rewards from temporal signals, can still be applied effectively.
>
> Although Meta-World is a single benchmark, it contains diverse manipulation tasks. In these tasks, prior methods fail in distinct ways: some cannot capture local progress, while others overestimate rewards for failed behaviors, leading to unstable performance across tasks. TimeRewarder consistently provides dense rewards that reflect temporal progress, resulting in more stable and reliable performance across all tested tasks. Based on these results, we believe that TimeRewarder effectively addresses core challenges in reward learning and could similarly provide benefits in other benchmarks where prior methods encounter comparable issues.
>
> # Question2: Interval sampling
> In practice, we first sample a video, then sample a temporal offset $\Delta$ from an inverse-probability distribution, which covers the full range from adjacent frames to near full-trajectory pairs. We then uniformly sample a starting frame $t \in [0, T-\Delta]$ and form the pair $(o_t, o_{t+\Delta})$. This effectively samples arbitrary frame pairs with varying temporal distances from the same trajectory.
>
> # Question3: Open-source
> Yes, we plan to do that.
>
> # Minor Weaknesses
> 1. Following the previous works, we mean unlabeled action-free videos from the third view. Thank you for pointing that out, and we will make it clearer in the revision.
> 2. We thank the reviewer for the suggestion. Our intention was to convey that they are relatively lightweight compared to dense reward design. We agree that such signals are not always easy to obtain, and will correct the claim in the revision.
> 3. Yes, $\Phi^{-1}$ is the vector-to-scaler translation, and we will clarify it in the revision.

---

> > ### Author Rebuttal · Reviewer_KVj4 · 2026-04-01
> >
> > I thank the authors for the clarifications, my concerns regarding the clarity of the paper are fully resolved. I acknowledge that I have read the other reviews and their corresponding responses.
> >
> > I think the paper is interesting and has novel contributions. I will keep my score for now as my concern regarding the limited evaluation benchmark (MetaWorld) is not easily addressable in the short rebuttal period. Though MetaWorld is an important simulated benchmark, I don't think it provides enough evidence for the claims in the paper, and I think an additional environment would strengthen the paper and its contribution.

---

> > > ### Author Response · Authors · 2026-04-08
> > >
> > > Thank you for your continued engagement, and we are very glad to hear that your concerns regarding the clarity of the paper have been fully resolved and that you find our contributions novel and interesting.
> > >
> > > We completely agree with your constructive feedback: while MetaWorld is a standard benchmark for robotic manipulation, validating our method on an additional, distinct simulator significantly strengthens the paper and proves the generality of our claims. Despite the tight time constraints of the short rebuttal period, we worked around the clock to set up and run evaluations on a completely different environment: the ManiSkill benchmark.
> > >
> > > Specifically, we have completed the training and evaluation for the PushCube-v1 task. We maintained a strictly identical setup to our Meta-World experiments: we used the first 100 trajectories from the official ManiSkill demonstrations for all the reward models (including TimeRewarder, VIP, Progressor,and Rank2Reward) and ILFO (OT, GAIfO) methods, and subsequently trained the downstream RL policies based purely on these rewards.
> > >
> > > After 110k environment steps, the results are as follows (please refer to Figure 3 in https://openreview.net/pdf?id=7yL1wteXNr for the detailed curve):
> > >
> > > | Environment | Method              | Success Rate (mean ± std, 8 seeds) |
> > > |-------------|---------------------|--------------------|
> > > | PushCube-v1 | **Environment reward** | **0.89 ± 0.13**    |
> > > |             | OT                  | 0.06 ± 0.07        |
> > > |             | GAIfO               | 0.03 ± 0.07        |
> > > |             |  **TimeRewarder**       | **0.86 ± 0.10**        |
> > > |             | VIP                 | 0.54 ± 0.27        |
> > > |             | Rank2Reward         | 0.00 ± 0.00        |
> > > |             | PROGRESSOR          | 0.75 ± 0.17        |
> > >
> > >
> > > Overall, the study evaluates a central hypothesis: progress can be reliably extracted from frame-wise temporal distances. TimeRewarder’s strong performance on ManiSkill provides further evidence that this signal generalizes across different simulator dynamics and visual rendering.
> > >
> > > We will include this complete ManiSkill evaluation (along with additional tasks beyond PushCube as time permits) in the final camera-ready version of the paper.
> > >
> > > Once again, we deeply appreciate your patience, your rigorous feedback, and your recognition of our work's novelty. We hope that this new, strong empirical evidence on ManiSkill adequately addresses your concern regarding the evaluation scope.

---

### Official Review · Reviewer_mQoy · 2026-03-06

**Soundness:** 3
**Presentation:** 2
**Significance:** 3
**Originality:** 3
**Overall Recommendation:** 4
**Confidence:** 3

**Summary:**

The paper proposes *TimeRewarder*, a method for automatically learning dense reward functions for reinforcement learning from passive video demonstrations without action annotations.
*TimeRewarder* jointly maps the adjacent observed frames into a discretized two-hot space spanning the interval [-1, 1].
Supervision in Eq. (5) explicitly trains the model using the relative frame distances and each frame's distance to the goal frame.
The learned estimator provides step‑wise rewards that guide RL agents on ten Meta‑World manipulation tasks, with experimental results demonstrating effectiveness across diverse environments.

**Compliance With Llm Reviewing Policy:**

Affirmed.

**Final Justification:**

I thank the authors for their clear and constructive rebuttal. After carefully considering both the paper and the response, I find that my initial concerns have been adequately addressed.

The paper presents TimeRewarder, a practical and scalable framework for deriving dense rewards from passive video data. It reduces reliance on manual reward engineering and demonstrates consistent empirical gains across diverse manipulation tasks, with a simple design that is easy to integrate into existing RL pipelines.

Importantly, the rebuttal directly addressed my main concerns, particularly around methodological justification and empirical coverage. These clarifications have positively changed my assessment. As a result, I have increased my score from 3 to 4.

Overall, I believe this work makes a solid contribution, with a good balance between simplicity and effectiveness.

**Key Questions For Authors:**

Q1. Does *TimeRewarder* explicitly use both frame order and frame-goal distance simultaneously through its two-hot distribution output space, in a way that clearly distinguishes it from existing progress-estimation-based RL algorithms?

Q2. It is easy to compute $\hat{d}$ through Eq. (3) by directly using $u$ and $v$ in $\hat{y} _ {uv}  = F _ \theta$ without knowing $\Phi^{-1}$. In other words, is $\Phi^{-1}$ redundant? If not, how can one prove that $\Phi$ admits an (approximate) inverse in this setting?

Q3. In the original Rank2Reward paper, the model outputs rewards via a progress-correlated ranking function $\hat{u}_\theta (s)$ that ranks the current frame to a task progress ratio. Why, in Experiment Question 1, is Rank2Reward excluded because it “lacks a progress function”? In addition, why in Experiment Question 2, a pairwise ordering reward instead of the original reward of Rank2Reward is reported?

**Limitations:**

Temporal distance prediction may struggle on tasks with highly stochastic dynamics and back-and-forth settings where frame ordering is ambiguous; no analysis is provided for such scenarios.

**Strengths And Weaknesses:**

**Strengths**:

1. Demonstrates a practical, scalable way to obtain dense rewards from readily available passive videos, reducing manual reward engineering.
2. Shows empirical gains across a wide variety of manipulation tasks.
3. The method is simple to implement and could be integrated into existing RL pipelines with minimal overhead.

**Weaknesses**:

1. **Main Contribution Unclear**
   The differences and advantages of the proposed method relative to prior work remain somewhat unclear. In Eq. (6), the distance tends to fluctuate around a small fixed value because $o_t$ and $o_{t+1}$ are temporally adjacent. As a result, the effectiveness likely stems primarily from frame ordering and frame-to-goal distance, both of which have been explored extensively in similar prior works.

2. **Unclear Mapping from Relative Distance to Absolute Progress**
   The method emphasizes the temporal distance between ordered frame pairs, but it is unclear how a relative distance can reliably convey absolute task progress information.

3. **Flaws in the Pipeline Logic**
   $\Phi$ takes the distance between frames $u$ and $v$ as input and outputs a two-hot distribution. A key issue is that $d_{uv}$ is constant in most cases, since adjacent frames are preferentially sampled as input to $\Phi$. This means a constant input distance is mapped to varying outputs to reflect task completion progress.  Such a process renders $\Phi$ a non-deterministic function and not fully learnable, even with a neural network.

4. **Motivation Uncovered by the Proposed Method**
   The model relies solely on adjacent frame pairs from the current observation, without incorporating any historical information.
   The resulting learned states are likely to become confused in more general and complex environments.
   For example, as noted in the limitations, the proxy reward fails in back-and-forth scenarios (e.g., drawer pull-out/check/push-in).
   The justification in Section 4.3 overlooks these general RL control scenarios.
   A clearer description of the contexts/scenarios where the method is expected to apply would be helpful and should be included in the introduction and method sections.

5. **Lack of Experimental Details**
   Figure 4 is not sufficiently informative, and the results are difficult to compare directly. Rewards from different algorithms are not properly aligned for fair comparison.  Moreover, it is surprising that Rank2Reward fails to predict the order of adjacent frames in most of these simple scenarios shown in Figure 4. Additional explanation and evidence for these failure cases, along with details on baseline implementation/evaluation to reproduce the reported results, would be valuable.

---

> ### Author Rebuttal · Authors · 2026-03-28
>
> # Weakness 1 & Weakness2 & Question1: Advantages to prior work
> We thank the reviewer for these questions. They seem to stem from a misunderstanding of how TimeRewarder is trained and what signal it learns, and we will clarify this in the revision.
>
> TimeRewarder predicts signed temporal distances between arbitrary frames, which are consistent with absolute task progress. In particular, the absolute progress of $o_t$ can be expressed as the accumulation of relative pairwise distances along the trajectory, i.e., the sum of distances from $o_{i-1}$ to $o_i$ for $i=2,\dots,t$. Therefore, although in RL, the supervision is pairwise, these relative distances fully determine a globally consistent notion of progress, rather than merely local or order-only information.
>
> This also clarifies the distinction from prior work. Rank2Reward captures only ordering without a distance measure, while PROGRESSOR models only forward distance to the goal with a more complex objective. VIP learns an implicit embedding to measure distance to goal; however, as we show in our **rebuttal to Reviewer 1dpV**, its objective is ill-posed and does not ensure stable optimization. Additionally, both VIP and PROGRESSOR require a goal image as input to the reward model, introducing extra assumptions and dependencies on goal specification.
>
> In contrast, TimeRewarder directly models a globally consistent notion of temporal progress across all frame pairs, enabling it to capture both fine-grained transitions and long-horizon structure within a single objective. Importantly, this formulation already jointly encodes ordering (sign) and distance (magnitude) in a **unified target space**, rather than treating them as separate signals. The two-hot output is a practical parameterization of this idea, not the core contribution.
>
> Regarding Eq. (6), although adjacent frames may yield small local distances, training is not restricted to such pairs. By sampling frame pairs with varying temporal gaps, the model learns consistency across short- and long-range relationships, avoiding reliance on noisy local differences.
>
> Overall, by **directly modeling signed temporal distances with global consistency**, TimeRewarder provides a well-posed and stable objective that naturally encodes both ordering and magnitude of progress, enabling reliable recovery of absolute task progress without goal conditioning.
>
> # Weakness 3 & Question2: About the scaler-to-vector function
>
> We clarify that the concern does not reflect an actual flaw, and we will make this point clearer in the revision. The mapping from scalar temporal distance to the two-hot output is **not learned** and is **fully deterministic**, so there is no ambiguity. Specifically, the distribution map $\Phi$ is a **fixed, invertible transformation** (see the cited method in line 194). This ensures a one-to-one correspondence between the scalar signed temporal distance and its vector representation.
>
> # Weakness5 & Question3: Rank2Reward visualization
>
> We thank the reviewer for pointing this out. We indeed discovered an error in the original visualization Figure 4: Rank2Reward should compute rewards based on the initial frame and the current frame, instead of the neighbor frames. We will correct the figures and related descriptions in the revised version.
>
> After the fix, the output of Rank2Reward is a natural progress estimation, as the visualization of other methods. Rank2Reward behaves more as expected, with successful trajectories showing rising rewards. For the window-open task, similar to VIP, reward differences between success and failure trajectories remain small, limiting its ability to fully distinguish outcomes.
>
> These corrections do not affect the main conclusions of our paper. We sincerely appreciate the reviewer’s careful observation of this baseline, which helps clarify and better present the contributions of our work.
>
> # Weakness4: Limitations  in back-and-forth scenarios
>
> As noted in our limitations, TimeRewarder assumes roughly monotonic progress—similar to prior progress-estimation-based methods—and operates on adjacent frame pairs without explicitly incorporating historical information. This can lead to ambiguous or unstable rewards in back-and-forth scenarios, as the reviewer noted.
>
> At the same time, this design allows us to study how to efficiently leverage temporal information as an unsupervised signal under a simple and scalable formulation. In more complex environments, the method can still be applied locally (e.g., combined with hierarchical mechanisms), since segments of approximately monotonic progress remain exploitable.
>
> We agree that incorporating history is an important direction, and as mentioned in our limitations, extending the method with memory-augmented or hierarchical progress models is a promising avenue for future work. We will also make the monotonic progress assumption and applicable scenarios more explicit in Section 4.3, as well as in the Introduction and Method sections.

---

> > ### Author Rebuttal · Reviewer_mQoy · 2026-04-03
> >
> > Thank you for the detailed rebuttal. I have carefully examined the responses to my comments as well as those of the other reviewers. The clarifications and newly conducted experiments have satisfactorily addressed my concerns to a reasonable extent.

---

> > > ### Author Response · Authors · 2026-04-03
> > >
> > > Thank you for acknowledging our rebuttal and for carefully reviewing both our responses and those of other reviewers. We are glad that the clarifications and additional experiments addressed your concerns. We will make sure that the explanations and applicable scenarios for TimeRewarder, including its assumptions and limitations, are clearly highlighted in the revised version to aid understanding for all readers.
> > >
> > > We would be grateful if your current assessment could be reflected in the score as appropriate.

---

### Official Review · Reviewer_QEUo · 2026-03-09

**Soundness:** 2
**Presentation:** 3
**Significance:** 2
**Originality:** 2
**Overall Recommendation:** 5
**Confidence:** 4

**Summary:**

The paper proposes TimeRewarder, a simple method for learning dense rewards from video demonstrations by predicting the temporal distance between frame pairs, which serves as a proxy for task progress. A progress model is trained with (1) negative sampling; (2) weighted pair sampling; and (3) two-hot discretization. Then, the predicted progress between adjacent frames is used as a stepwise dense reward in addition to the task-sparse reward to guide DrQ-v2's online policy learning. Experiments on 10 MetaWorld manipulation tasks demonstrate improved success rates and sample efficiency compared with prior reward-learning baselines. Experiments also analyze generalization to OOD experts, suboptimality identifiability, and cross-domain ability. In the end, the ablation study show usefulness of three training tricks together with the possibility of algorithm variants.

**Compliance With Llm Reviewing Policy:**

Affirmed.

**Final Justification:**

I appreciate TimeRewarder, as a simple method, empirically performs well over competitive baselines on the standard MetaWorld benchmark. Through rebuttal discussions with all reviewers, the authors have well differentiated TimeRewarder from other closely-related methods, e.g., Rank2Reward, Progressor, and VIP. Especially, the representation comparison between TimeRewarder and VIP sheds insight into why it works beyond "it just works". My own concerns about the mathematical rigor in Section 4.3 and the fair comparison with more recent ILfO baselines have also been well addressed, which should now interest a broader community in terms of learning dense rewards. Finally, limited empirical evaluation beyond the single benchmark and the "roughly monotonic progress" assumption may influence this work's impact. Balancing these reasons, my final score is 5 (3$\rightarrow$5), and I am generally positive about this work.

**Key Questions For Authors:**

My major concerns are:

1. Could the author go beyond Figures 3 and 4 to provide further insights into the feasibility and effectiveness of TimeRewarder, given the mentioned progress-based related work?
2. Could the author design fair comparisons with Temporal OT and ORCA to validate its performance superiority?

**Limitations:**

See weaknesses.

**Strengths And Weaknesses:**

***Strengths***:

1. This paper is well-written and easy to follow with good visualizations and diagrams.
2. The research question is important, and the problem setup is clear and focused.
3. The proposed method, TimeRewarder, is simple and intuitive, which yields strong performance in the default setting compared with the considered baselines.
4. Codes are provided in the supplementary material, which facilitates reproduction.
---
***Weaknesses:***

1. Despite the strong empirical performance, ___deeper insights into why such a simple method works online remain limited___ for further development along this line of research. In terms of the considered progress-based baselines, VIP, GVL, Rank2Reward, and PROGRESSOR all rely fundamentally on and exploit temporal distance signals encoded in the collected expert demonstrations to reward online agents. Why do their performances mismatch TimeRewarder?
    1. For example, the authors argue that the VIP objective is hard to optimize. This is a good starting point for a deeper investigation.
    2. In Figure 7, order prediction resembles GVL training. Does this prediction add complexity or noise so that it results in worse reward signals?
    3. And when the authors say Rank2Reward and PROGRESSOR suffer OOD issues, why does TimeReward not encounter such issues, given 100 demos and then bypassing online refinement?

    Without differentiating from them in depth, the significance and novelty of this work diminish.

2. ___The theoretical justification is rather vague.___ From my understanding, the authors want to convey in Section 4.3: if the potentials are defined as the expert-demonstrated temporal shortest path to a goal, then potential-shaped rewards $F_\theta$ can be directly modeled by temporal distances among expert states. To get this message across (if I am correct), Section 4.3 needs more effort to improve clarity, definitions, and math notations. For example, I have to assume Equation (8) takes expectation over all demos, and think that the potential definition should not depend on the timestep and demonstration path. Moreover, under this potential definition, the policy invariance guarantee may be fragile, as potentials depend largely on expert demos and $F_\theta$ may generalize poorly, during online training, to any unseen observations beyond expert-induced state distribution.

3. ___Baselines representing ILfO are outdated and weak.___ For example, Temporal OT [1] identifies the drawbacks of OT rewards and shows performance superiority over GAIfO, OT, and ADS on MetaWorld with DrQ-v2, given two demos. They also show positive scaling with the number of experts (Figure 5) and results with pixel-based inputs (Figure 6). For example, in the basketball environment, TemporalOT (pixel inputs; without sparse reward addition; given two demos) has much better performance at 2e5 env steps compared to TimeRewarder in a similar setting in Figure 14. Later on, ORCA [2] further identifies issues with TemporalOT, improves on top of that, and gets better results (e.g., Figures 5/6) with enough insights and depth.
4. ___Some empirical results are not as strong as claimed.___ First, comparing Figure 14 and Figure 5, it seems that sparse reward improves TimeRewarder a lot, and removing it eclipses performance superiority over baselines. Second, cross-domain results seem to suffer notable variance and instability, which undermines the claim “TimeRewarder showed successful cross-domain learning ability…”
---
***Minor issues:***

1. Could the authors provide details on the inverse map $\Phi^{-1}$? How does TimeRewarder aggregate the predicted distance bins?
---
***References:***

[1] Robot Policy Learning with Temporal Optimal Transport Reward (Fu et al., 2024)

[2] Imitation Learning from a Single Temporally Misaligned Video (Huey et al., 2025)

---

> ### Author Rebuttal · Authors · 2026-03-27
>
> # Weakness1 & Question1: Insights into why TimeRewarder works.
>
> We thank the reviewer for their suggestions and address the three points below:
> 1.  We provide a theoretical analysis showing that the VIP objective is hard to optimize: its loss is unbounded below and admits degenerate solutions with unbounded representations. Due to space limits, please refer to our **rebuttal to Reviewer 1dpV** for details.
> 2.  We attribute the weaker performance of order prediction methods to: (1) permutation classification ignores temporal deviation magnitude; (2) frame downsampling limits pairwise supervision. This leads to less effective supervision than TimeRewarder, which exploits all frame pairs with signed distances.
> 3.  TimeRewarder learns signed distances over all frame pairs, providing richer and more consistent supervision than ordering-only (Rank2Reward) or forward-only (PROGRESSOR), improving generalization and reducing OOD issues.
>
> # Weakness3 & Question2: More ILfO baselines.
> We thank the reviewer for mentioning these recent ILfO baselines. We will include both TemporalOT and ORCA in the revision with fair comparisons. We have implemented them on four tasks; results at 2e5 steps are shown in the table (mean ± std, 8 seeds).
> |Version|Method|button-press-topdown|door-open|drawer-open|basketball|
> |:-|:-|:-:|:-:|:-:|:-:|
> |**default** (Figure 5)|OT|0.00±0.00|0.00±0.00|0.00±0.00|0.00±0.00|
> ||ADS|0.41±0.34|0.00±0.00|0.49±0.35|*0.04±0.10*|
> ||TemporalOT|0.00±0.00|0.00±0.00|0.00±0.00|0.00±0.00|
> ||ORCA|*0.75±0.43*|*0.01±0.03*|*0.84±0.32*|0.00±0.00|
> ||TimeRewarder|**1.00±0.00**|**1.00±0.00**|**1.00±0.00**|**1.00±0.00**|
> |**w/o sparse reward** (Figure 14)|OT|0.01±0.03|*0.01±0.03*|0.00±0.00|0.00±0.00|
> ||ADS|*0.54±0.31*|0.00±0.00|*0.29±0.34*|**0.08±0.16**|
> ||TemporalOT|0.00±0.00|0.00±0.00|0.00±0.00|0.00±0.00|
> ||ORCA|0.23±0.19|0.00±0.00|0.00±0.00|0.00±0.00|
> ||TimeRewarder|**1.00±0.00**|**0.80±0.12**|**1.00±0.00**|*0.01±0.03*|
>
> In our setting, TemporalOT does not outperform ADS, consistent with our observation that OT-based methods rely on near-identical initial states; the masking mechanism in TemporalOT further exacerbates this, leading to near-zero success when trajectories differ. This matches observations in ORCA.
>
> ORCA can outperform ADS, but its max-based alignment allows frame skipping without penalty, and its multiplicative structure penalizes long-horizon matches, resulting in strong task-dependent variance, as reflected in the table.
>
> In contrast, TimeRewarder learns signed temporal distances over all frame pairs, avoiding both compounding errors and frame-skipping issues, and achieves more stable performance across tasks.
>
> # Weakness2: Clarity in theoretical justification
> We thank the reviewer for the careful reading of Section 4.3 and for correctly interpreting our intent.
> 1. **Clarification of the formulation.** We agree that the presentation can be clearer. Eq. (8) should explicitly take expectation over demonstrations, i.e., $\mathbb{E}_{\tau \sim \mathcal{D}_E}[\cdot]$, which we will fix. We also agree with your critique regarding the potential function's dependence. In standard Potential-Based Reward Shaping (PBRS, Ng et al., 1999), the potential function $\Psi(s)$ must strictly be a function of the state space. In the revision, we will rigorously redefine $\Psi(s)$ as a mapping solely from the state $s$, utilizing a distance metric to the goal, stripping away any misleading time or path-dependent notation.
> 2. **On policy invariance.**  In TimeRewarder, the potential is learned only from expert data under the assumption that it reflects optimal behavior. We do not perform online refinement of the reward model during RL, so the policy invariance guarantee is unaffected even when the agent encounters imperfect out-of-distribution (OOD) trajectories.
> 3. **On generalization beyond expert data.** The concern instead lies in practical generalization. If the agent visits states outside the expert distribution, the learned potential may provide weaker guidance. In our method, since temporal distances are modeled with continuous function approximators (neural networks), the potential varies smoothly beyond the training data. While the reward signal may degrade in such regions, it does not introduce inconsistency, and empirically we observe stable learning.
>
> # Weakness4: About empirical results
> 1. **On sparse reward.** We agree that sparse reward significantly improves performance. Our goal is not to replace it, but to provide a complementary dense signal. We will clarify this to avoid overclaiming.
> 2. **On cross-domain performance.**  We acknowledge the higher variance in cross-domain results due to domain gaps. While not yet fully stable, TimeRewarder suggests a promising transfer of temporal signals. We will revise the wording accordingly.
>
> # Minor issue:
> The distribution map $\Phi$ is a fixed, invertible transformation; see the cited method in line 194 for details.

---

> > ### Author Rebuttal · Reviewer_QEUo · 2026-04-02
> >
> > I confirm that I have read the author's rebuttal and other reviewers' comments. I thank the authors for responding to my review and conducting extra experiments. Most of my key concerns are addressed, but before I increase my score, I'd like to further verify the following:
> >
> > 1. The extra experiments with TemporalOT and ORCA look good, and I recommend including them in the paper revision. I went through Appendix A.4.2 again to remind myself of how demonstrations are collected. I want to double-check if all ILfO baselines were trained/compared on the same 100 demonstrations used to train TimeRewarder, e.g., diverse start-goal pairs? If yes, in particular, how did you aggregate the relabeled rewards computed from 100 demos for TemporalOT? I assume that you chose the best set of relabeled rewards that induces the highest relabeled return? In addition, I suggest that the authors mention this data collection protocol in the main body, as I found this detail very important when comparing TimeRewarder with ILfO baselines.
> >
> > 2. Although I was not concerned too much by Section 4.3 in my original review, after reading other reviews, I agree that the limitation about "back-and-forth motions" and state/observation revisitations should be clearly mentioned somewhere in the main body in order to properly scope the paper's contribution. I think Section 4.3 is a good place to formally do this. Could the authors write down the rigorous redefinition of Equation (8) promised in the rebuttal? The authors can either choose to enforce the "roughly monotonic progress" assumption to make the formulation easier, or give a more general definition that can handle state revisitations. An example of observation revisitation looks like: a demo like ($o_0, o_1, o_2, o_3, o_1, g$), where the first $o_1$ may describe "hand position upon opening the drawer to place an object" and the second $o_1$ may describe "hand position upon closing the drawer after putting the object inside". Ideally, we want the temporal distance between $o_1$ and $g$ to be 4, instead of 1 or (1+4)/2=2.5.
> >
> > ---
> >
> > Minor comment: Two other reviewers were also confused by $\Phi^{-1}$. I think this confusion could have been largely avoided if the authors simply indicated that $\Phi^{-1}$ is the "Gumbel-Max trick".
> >
> > ---
> > **Update:** I have read the authors' additional rebuttal comment. The revised Section 4.3 looks better than before, and I believe the authors will scope the paper's contribution by explicitly discussing the "back-and-forth motions" limitation to avoid confusion and misunderstanding. My concerns have been fully addressed. I also like the analysis of TimeRewarder and VIP representations. I therefore increase my score to 5.
> >
> > **Caveat:** According to the rule, an anonymous link should only contain figures and tables with captions. Although I am fine with the current link and suggest other reviewers and AC not overreacting to it, if possible, the authors should properly deal with it, e.g., include the condensed Section 4.3 revision into the rebuttal comment directly.

---

> > > ### Author Response · Authors · 2026-04-04
> > >
> > > We sincerely thank you for engaging with our rebuttal and for the highly constructive follow-up question. We hope these clarifications resolve the remaining concerns and make the contributions and comparisons clear.
> > >
> > > **1. On TemporalOT reward aggregation**.  Yes, all ILfO baselines are trained/compared on the same 100 demonstrations.
> > > For TemporalOT, we follow the OT/ADS original implementation: for each agent trajectory, rewards are computed w.r.t. the most aligned demonstration (i.e., the one yielding the highest matching score).
> > >
> > > We agree that this detail is important for fair comparison and will clarify it in the main text.
> > >
> > > **2. On state revisitation**. We completely agree that clearly scoping the paper's contribution by explicitly discussing the "back-and-forth motions" limitation and observation revisitation greatly strengthens the paper.
> > >
> > > To address your feedback, in the revised Section 4.3, we proceed to present a central topic: the rigorous mathematical formulation of temporal distance as a valid shaping potential under a Markovian state assumption, followed by an explicit discussion of how this translates to partial observations and visual aliasing in practice:
> > >
> > > ----
> > > **Revised Section4.3**
> > >
> > > We provide a theoretical justification for our motivation that the task progress in expert videos can be formalized in terms of temporal distance.
> > >
> > > Consider a fully observable Markov Decision Process (MDP), where the true state $s \in \mathcal{S}$ captures all task-relevant information (e.g., object poses, velocities, and gripper status). Given a goal state $s_g$, we define a sparse reward with a per-step penalty:
> > > $r(s) = -1$ if $s \neq s_g$, and $r(s) = 0$ if $s = s_g$
> > >
> > > and assume deterministic transitions $s' = f(s, a)$.
> > >
> > > Under this setting, the optimal value function satisfies the Bellman optimality equation:
> > >
> > > $$
> > > \mathcal V_{\gamma}^{\*}(s) = r(s) + \gamma \max_{a} \mathcal V_{\gamma}^{\*}(f(s, a)).
> > > $$
> > >
> > > For any expert trajectory $\tau^e = (s_1^e, \dots, s_T^e)$ generated by the optimal policy, the value of each visited state admits a closed-form expression:
> > >
> > > $$
> > > \mathcal V_{\gamma}^{\*}(s_t^e) = -\sum_{k=t}^{T-1} \gamma^{k-t}, \mathcal V^{\*}(s_g) = 0.
> > > $$
> > >
> > > Therefore, $\mathcal V^*(s)$ is a monotonic transformation of the remaining time-to-go $T - t$.
> > >
> > > This naturally suggests defining a potential function:
> > > $$
> > > V(s_t^e) = - \mathcal V_{\gamma=1}^*(s_t^e) = T - t,
> > > $$
> > > which maps each state to its temporal distance to the goal.
> > >
> > > **Connection to visual RL.** To bridge the above formulation with visual observations, we assume the underlying state can be approximately reconstructed from observations, i.e., $s \approx \phi(o)$. This requires that each observation uniquely identify the task phase, thereby ruling out cyclic trajectories that revisit visually indistinguishable states. In practice, however, visual RL operates in a Partially Observable MDP (POMDP), where this assumption can be violated due to \emph{visual aliasing}, particularly in back-and-forth motions. In such cases, disambiguating different visits would require history-dependent information (e.g., a loop counter or velocity), which is not available from single-frame observations.
> > >
> > > For example, consider a trajectory $(o_0, o_1, o_2, o_3, o_1, o_g)$, where the first occurrence of $o_1$ corresponds to the hand reaching to open a drawer, and the second corresponds to retracting after placing an object. In the underlying state space, these correspond to distinct states $s_1$ and $s_4$, with true temporal distances of $4$ and $1$ steps to the goal, respectively, thus preserving monotonic progress. However, when restricted to single-frame observations, both states collapse to the same $o_1$, leading a learned temporal distance model $F_\theta$ to produce an averaged estimate (e.g., $2.5$), which may locally violate monotonicity.
> > >
> > > To resolve this in practical deployments where back-and-forth motions are frequent, TimeRewarder can be extended by replacing single-frame inputs with observation histories (e.g., concatenating a time window of frames $(o_{t-k}, \dots, o_t)$). By incorporating history, the input once again approximates the true Markovian state vector, effectively disambiguating the temporally distinct visits to $o_1$ and recovering the accurate progress distances of 4 and 1.
> > >
> > > ----
> > > We will incorporate all of these updates into the final camera-ready version. Thank you again for helping us improve the clarity and rigor of our paper!
> > >
> > > (Following the reviewer’s suggestion and to comply with the anonymity guidelines, we have moved the relevant content directly into this rebuttal for clarity and ease of reference.)

---

### Official Review · Reviewer_1dpV · 2026-03-13

**Soundness:** 4
**Presentation:** 3
**Significance:** 3
**Originality:** 2
**Overall Recommendation:** 5
**Confidence:** 5

**Summary:**

The authors introduce TimeRewarder, a method to learn a dense reward function from passive, action-free observations (e.g. videos). The crux of the method is to fit an estimate of the (discretized) temporal distance between frames with asymmetric sampling by reversing examples in time. The method shows improvements over similar methods in MetaWorld.

**Compliance With Llm Reviewing Policy:**

Affirmed.

**Final Justification:**

The authors introduce TimeRewarder, a method to learn a dense reward function from passive, action-free observations (e.g. videos).

The significance of the method is that the method is simple, and makes principled improvements over prior related methods like VIP and PROGRESSOR.

While I don't think this method is particularly original -- all of these individual components to the method have been implemented in some place or another -- it is nice to see key design decisions come together in a simple algorithm that works for an adequate benchmark.

The authors' rebuttals also helped answer why their method performs better than similar methods which raised my score from 2 --> 5. There is now direct empirical evidence for the shortcomings in prior methods that they identified in the rebuttals, and also helps readers not only learn a new method but also understand failure modes of prior methods and why the new method works. Overall, I the significance and quality of the work was made clear to me during rebuttals, and I support acceptance.

**Key Questions For Authors:**

- How does the method mathematically compare with prior methods, especially VIP?
- Why is there an improvement over prior mathematically-similar methods? If possible, is there an analysis of why the most stark improvement occurs on specific tasks (e.g. Basketball vs. Plate slide)?

**Limitations:**

yes

**Strengths And Weaknesses:**

Strengths:

- Learning from action-free expert data is an important problem, and specifying a dense reward from videos offers a way to transfer expert knowledge directly into existing RL frameworks.

- The frame-wise temporal distance formulation is simple. The implicit negatives from the antisymmetric parameterization of the logit is a good touch, though previously implemented elsewhere.

- Evaluations done across the Metaworld suite with modest improvements over other methods.

- The experiments mixing Metaworld and human videos is very cool, I am surprised by the strength in performance relative to 1 MW and 20 Human.

Weaknesses:

Major: conceptual/theoretical argument of improved performance over prior methods.
- The VIP objective (Ma et al. 2022) and Time Rewarder seem very mathematically similar. Note that the Value function in 4.3 is identical to the (negative) Value function in VIP.

The difference, from what I can tell, is rewarding a forward temporal distance, rather than a decrease in the temporal distance to a desired goal.

``However, we found [VIP] difficult to optimize reliably'' is not sufficient -- given the similarities between TimeRewarder and recent methods, it makes it important to present an empirical *and* conceptual understanding of why TimeRewarder shows stronger or different performance. I believe this is necessary to show significance and originality of the work.

Minor:
- The weighted pair sampling scheme does not seem like a natural construction. Does sampling $\Delta$ from a geometric distribution work?

- In 3.2, it seems important to note that reward shaping from TimeRewarder should typically leave the optimal policy invariant. It is not obvious from the discussion.

- Be careful about value as defined in Section 4.3 vs. true value used to train the method. They are different, and could be made clearer w/ a short note.

- Fig. 4, while communicating an important point, is hard to understand. Mixing reward and value y-axes is confusing. Maybe turn everything to values / just integrate the reward for Environment, Rand2Reward, and PROGRESSOR?

**If the Weaknesses (particularly more careful theoretical analysis in service of comparing and conceptually presenting the difference between TimeRewarder and prior methods, especially VIP/other methods that rely on temporal distances) are addressed, I am willing to raise my score.**

---

> ### Author Rebuttal · Authors · 2026-03-25
>
> # Major Weakness & Questions: mathematical difference from VIP and other prior methods
> We appreciate the reviewer’s recognition that temporal distance is a strong supervisory signal for learning reward from unlabeled videos. While prior works also exploit this idea, effectively modeling it remains challenging. TimeRewarder provides a simple and scalable mechanism to leverage temporal progress in learning rewards.
>
> Although the value functions may appear similar, the training objective of TimeRewarder is fundamentally different, leading to distinct and more reliable optimization behavior.
>
> **1. Theoretical Analysis of VIP's Objective.**
>
> Mathematically, the training objective of TimeRewarder and VIP is fundamentally different. We note that the objective equation in the VIP paper, Eq.(6), is inconsistent with its pseudocode in Appendix D.3. Following the pseudocode (consistent with the official codebase), the sign before $\gamma$ is reversed. Under this formulation, we show that the VIP loss has no lower bound and admits degenerate solutions with unbounded representations:
>
> In VIP Section 4.1, $\tilde\delta_g(o)=\mathbb I(o==g)-1$, so $\tilde\delta_g(o)=0$ if and only if $o=g$ instead of $o_{next}=g$. (VIP codebase: "this is always -1" in data_loaders.py.)
>
> Consider an $N+1$-state chain-like MDP with initial state $o_0$ and goal $g=o_N$. For a full-trajectory batch $\{o_0,o_1,\ldots,o_{N-1}\}$, all samples share the same $o_0$. Note that $V_{o_t}=-\|\phi(o_t)-\phi(g)\|$, and we have
>
> $$\mathcal{L}=(1-\gamma)(-V_{o_0})+\log\left(\frac{1}{N}\sum_{t=0}^{N-1}\exp\left(-(\tilde\delta_g(o)+\gamma V_{o_{t+1}}-V_{o_t})\right)\right)=(1-\gamma)\|\phi(o_0)-\phi(g)\|+\log\left(\frac{1}{N}\sum_{t=0}^{N-1}\exp\left(\gamma\|\phi(o_{t+1})-\phi(g)\|-\|\phi(o_t)-\phi(g)\|\right)\right)+1$$
>
> Let $\|\phi(o_t)-\phi(g)\|=c\frac{1-\gamma^{N-t}}{1-\gamma}$ ($c>0$), we have
>
> $$\gamma\|\phi(o_{t+1})-\phi(g)\|-\|\phi(o_t)-\phi(g)\|=c\frac{\gamma(1-\gamma^{N-(t+1)})-(1-\gamma^{N-1})}{1-\gamma}=c\frac{\gamma-1}{1-\gamma}=-c,$$
>
> Thus
>
> $$\mathcal{L}=(1-\gamma)c\frac{1-\gamma^N}{1-\gamma}+\log\left(\frac{1}{N}\sum_{t=0}^{N-1}\exp(-c)\right)+1=c(1-\gamma^N)-c+1=-\gamma^N c+1.$$
>
> Obviously $\lim_{c\rightarrow+\infty}\mathcal{L}=-\infty$.
>
> In practice,  we indeed observed that when effective representations encoding progress are learned, the solution approximately resembles the Bellman solution scaled by a constant, appearing to "converge." Yet this outcome is not guaranteed, which is why we noted in the paper that this objective is "difficult to optimize reliably."
>
> **2. Conceptual and Empirical Advantages of TimeRewarder.**
>
> Rank2Reward captures only relative ordering without distance information, while PROGRESSOR models only forward distance and relies on a more complex objective. Additionally, VIP and PROGRESSOR are goal-based methods that require an image goal as a reward model input during RL. By avoiding this requirement and using a self-consistent objective, TimeRewarder fully exploits temporal structure in videos as richer supervision, leading to more reliable optimization and consistently strong performance.
>
> **3. Analyze performance improvement on specific tasks?**
>
> In fact, TimeRewarder *does not* outperform prior methods by a large margin *only on specific tasks*. Rather, TimeRewarder achieves stable performance across diverse tasks, whereas prior methods fail on different tasks, resulting in substantially worse overall performance. Examining *drawer-open* in Fig. 5, Rank2Reward can achieve 100% success rate, PROGRESSOR achieves roughly 50%, while VIP fails completely with 0% success. Meanwhile, Appendix Fig. 12 shows that simply replacing VIP's vision encoder with ResNet yields strong performance on *drawer-open*, though this change harms other tasks such as *plate-slide*. This phenomenon aligns with our earlier analysis: VIP's optimization process is unstable in practice.
>
> Furthermore, the ablation study around line 414 analyzes why PROGRESSOR may fail on specific tasks. Removing TimeRewarder's Implicit Negative Sampling to match PROGRESSOR's forward-only temporal modeling causes performance drops on *stick-push* and *basketball*, where failed grasps are common and PROGRESSOR overestimates such failures as partial progress.
>
> # Minor Weaknesses
> W1: The training performance is insensitive to the specific sampling distribution; geometric sampling also works.
>
> W2&3: We thank the reviewer for these suggestions and will address them in the revision.
>
> W4: This visualization was chosen for clarity. We use "progress estimates" for all methods to better compare how each captures task progression. Since the rewards of Environment, Rank2Reward, and PROGRESSOR are mostly positive, plotting their cumulative reward directly yields mostly monotonically increasing curves. To highlight differences, we visualize their raw rewards. VIP and TimeRewarder use potential-based rewards, so we instead show their values as progress estimates.

---

> > ### Author Rebuttal · Reviewer_1dpV · 2026-04-02
> >
> > I believe my questions have been fully answered and have raised my score to a 4. I confirmed I read the other reviews.
> >
> > I thank the authors for their thorough response and engagement with my questions and concerns and strongly encourage authors to include these arguments and reasoning re: VIP and PROGRESSOR and **why their method avoids these pitfalls** within the main text.
> >
> > My remaining questions are about more thorough ablations of PROGRESSOR + VIP to clearly demonstrate the reason TimeRewarder has gains over prior methods. These experiments would raise my score to a 5, and make me more convinced about these differences.
> >
> > ------
> >
> > On the suggested ablations:
> >
> > (1) First, I was (incorrectly) under the assumption that VIP used a separate loss to train the representations such that it captures temporal distance. The authors are correct -- without additional regularization, a trivial solution is representation collapse. TimeRewarder clearly avoids this because the distance function learning is a supervised problem -- there's real signal from the actual indices of the frames that escapes this problem of bootstrapping learned representations to learn representations. I encourage adding this analysis into the main text.
> >
> > Something that would drive this point home and raise my score to a 5: your antisymmetric sampling implies a natural antisymmetric representation structure for y hat, e.g. y hat = Phi(psi(s') - psi(s)). Directly using this parameterization would enable you to make a comparison between the representations implicitly learned in your method and the representations learned by VIP w/ a similar-ish form, and show collapse (or the lack thereof).
> >
> > (2) Another good ablation would be implementing this antisymmetric negative sampling and/or parameterization in e.g. PROGRESSOR. If this improves PROGRESSOR, it would also help me/readers understand where the strengths of your method comes from. Figure 7 has me inclined to believe that adding such a structure would resolve the difference in performance between the performance of PROGRESSOR and TimeRewarder.

---

> > > ### Author Response · Authors · 2026-04-03
> > >
> > > We thank the reviewer for the detailed suggestions and for encouraging a deeper investigation. Based on the suggested ablations, we conducted the 2 experiments (Figures in https://openreview.net/pdf?id=7yL1wteXNr):
> > >
> > > (1) Following your advice, we modified TimeRewarder to explicitly learn representations (see Figure 1). Across both training and held-out expert demonstrations, TimeRewarder representations remain smooth and structured. Importantly, TimeRewarder can clearly distinguish successful and failed RL rollout trajectories: the representations for successful agent rollouts follow a coherent progression towards the goal, whereas failed rollouts deviate from the structure, reflecting task progress. VIP representations, by contrast, while capturing general trends on successful trajectories, are less smooth and fail to generalize as well to held-out demonstrations, and are less able to separate success from failure.
> > >
> > > Beyond the previously analyzed VIP objective issues, we further hypothesize that VIP’s reliance on a pre-defined goal image further limits generalization: in held-out trajectories, the observations can be far from the provided goal, which reduces the effectiveness of the learned representations. This limitation also applies to goal-based methods such as PROGRESSOR.
> > >
> > > (2) For PROGRESSOR, we implemented negative sampling, but we note that the architecture does not naturally support antisymmetric modeling. PROGRESSOR originally selects three consecutive frames $(o_1, o_2, o_3)$ and predicts the relative position of $o_2$ in $[0,1]$. With negative sampling, we can predict $o_1$ relative to $o_2$ and $o_3$, but the resulting range shifts to $[-T,0]$ rather than a symmetric $[-1,0]$. While this implementation is imperfect, it still improves performance on some tasks, such as window-close and drawer-open (see Figure 2). Tasks like basketball and stick-push still underperform compared to TimeRewarder, suggesting that PROGRESSOR’s structure is far from the natural antisymmetric representation achieved by TimeRewarder.
> > >
> > > ----
> > >
> > > We thank the reviewer again for the constructive suggestions. Taken together, these experiments provide a clearer understanding of why TimeRewarder outperforms prior baselines such as VIP and PROGRESSOR. These results complement our analysis of VIP’s objective and PROGRESSOR’s structural constraints, providing further insight into the sources of TimeRewarder’s gains. We fully agree that this analysis is insightful and will incorporate it into the main text along with the corresponding experimental results in the main paper or appendix.

---

### Decision · Program_Chairs · 2026-04-30

**Decision:**

Accept (spotlight)

**Comment:**

The paper introduces TimeRewarder, a framework designed to learn dense reward functions for Reinforcement Learning (RL) from passive, action-free video demonstrations. The core innovation involves training a progress model to estimate discretized temporal distances between frame pairs, effectively using time as a proxy for task completion. This approach employs techniques such as negative sampling, weighted pair sampling, and two-hot discretization to provide stepwise rewards that guide online policy learning.

All reviewers found the method to be well-motivated, simple to implement, and intuitive. The paper's strengths lie in its strong empirical performance and the practical value of learning from readily available expert videos without manual reward engineering. On the benchmarks, TimeRewarder demonstrated consistent improvements in success rates and sample efficiency over established baselines. Reviewers particularly highlighted the results involving the use of human videos to guide robotic tasks and the method's cross-domain learning capabilities. While initial concerns were raised regarding the technical novelty and the mathematical similarity to prior work, the authors' rebuttals were highly effective in differentiating TimeRewarder and providing empirical evidence for why it avoids the optimization failures or out-of-distribution (OOD) issues seen in previous methods. Some limitations remain, notably that the evaluation is primarily focused on a single simulated benchmark (MetaWorld), which may not fully capture the complexity of multi-object manipulation or highly stochastic environments. Additionally, some reviewers noted that the method might struggle with "back-and-forth" scenarios where temporal ordering is ambiguous.

However, the consensus is that the simplicity and effectiveness of the algorithm, combined with the clarifications provided during the rebuttal period, make this a significant contribution to the field of self-supervised reward design. Consequently, all reviewers recommend  acceptance.